# SPATIAL-DISE: A UNIFIED BENCHMARK FOR EVALUATING SPATIAL REASONING IN VISION-LANGUAGE MODELS

**Xinmiao Huang[1], Qisong He[1], Zhenglin Huang[1], Boxuan Wang[1], Zhuoyun Li[1],**
**Guangliang Cheng[1], Yi Dong[1,*], Xiaowei Huang[1]\***
[1]School of Computer Science & informatics, University of Liverpool

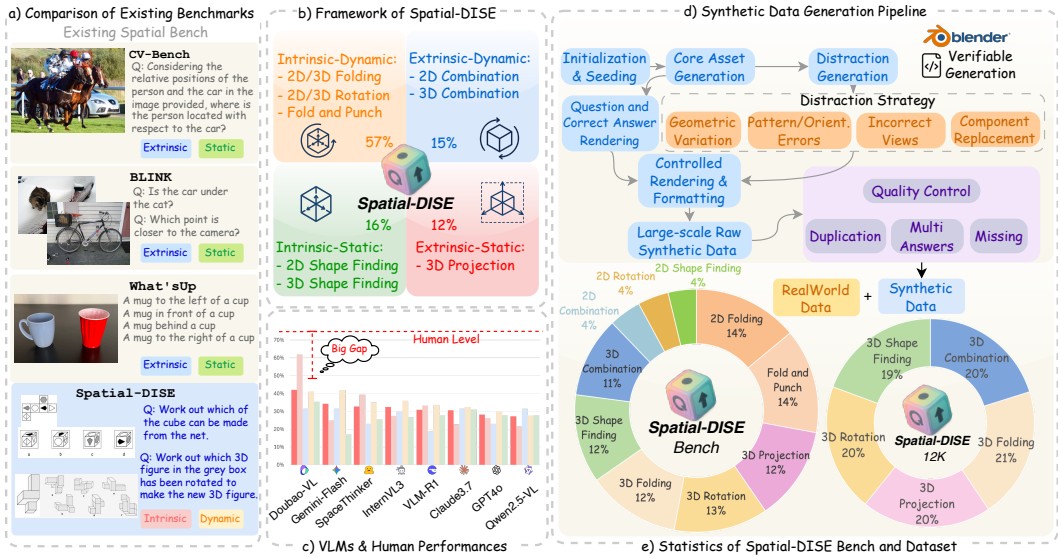

**Figure 1:** A Comprehensive Overview of the **Spatial-DISE** Framework, Generation Pipeline, and Benchmark Statistics. a) Comparison of examples from existing benchmarks, which primarily test general static reasoning, with cognition intrinsic-dynamic tasks from our Spatial-DISE benchmark. b) introduces the core DISE taxonomy, showing the four quadrants of spatial reasoning and their distribution in the 559-pair evaluation bench. c) presents evaluation results, showing a significant gap between model and human performance. d) details the synthetic data generation pipeline implemented in Blender, and e) provides a statistical breakdown of the task categories within both the Spatial-DISE Bench and the Spatial-DISE-12K.

## ABSTRACT

Spatial reasoning ability is crucial for Vision Language Models (VLMs) to support real-world applications in diverse domains including robotics, augmented reality, and autonomous navigation. Unfortunately, existing benchmarks are inadequate in assessing spatial reasoning ability, especially the *intrinsic-dynamic* spatial reasoning which is a fundamental aspect of human spatial cognition. In this paper, we propose a unified benchmark, **Spatial-DISE**, based on a cognitively grounded taxonomy that categorizes tasks into four fundamental quadrants: **I**ntrinsic-**S**tatic, Intrinsic-**D**ynamic, **E**xtrinsic-Static, and Extrinsic-Dynamic spatial reasoning. Moreover, to address the issue of data scarcity, we develop a scalable and automated pipeline to generate diverse and verifiable spatial reasoning questions, resulting in a new **Spatial-DISE** dataset that includes Spatial-DISE Bench (559 evaluation VQA pairs) and Spatial-DISE-12K (12K+ training VQA pairs). Our comprehensive evaluation across 32 state-of-the-art VLMs reveals that, current VLMs have a large and consistent gap to human competence, especially on multi-step multi-view spatial reasoning. Spatial-DISE offers a robust framework, valuable dataset, and clear direction for future research toward human-like spatial intelligence. Benchmark, dataset, and code are available at Spatial-DISE.

---

*Corresponding author: xiaowei.huang@liverpool.ac.uk, yi.dong@liverpool.ac.uk

# 1 INTRODUCTION

Recent advances in vision language models (VLMs) have demonstrated impressive capabilities in various tasks such as object detection (Li et al., 2022; Peng et al., 2023; Anil et al., 2025), scene caption (Alayrac et al., 2022; Chen et al., 2023; Li et al., 2023), and visual question answering (Wang et al., 2023; Anil et al., 2025; Alayrac et al., 2022). However, their capability for sophisticated *dynamic* spatial reasoning, a cornerstone of human cognition and a critical requirement for applications in robotics, augmented reality, and autonomous navigation, remains a significant limitation (Ramakrishnan et al., 2024) and largely under-evaluated.

Existing benchmarks for evaluating the spatial reasoning of VLMs have three major limitations. Firstly, current benchmarks **lack a systematic cognitive framework** for categorizing and evaluating different types of spatial reasoning abilities, leading to fragmented, unbalanced tasks that typically focus only on basic skills(Liu et al., 2023; Chen et al., 2024; Cheng et al., 2024). Consequently, there is a notable scarcity of benchmarks designed to evaluate deeper cognitive abilities. Secondly, current benchmarks are **limited in scope**, focusing predominantly on *static* spatial questions that do not require *multi-step dynamic* reasoning (Wang et al., 2024; Han et al., 2020). Consequently, crucial cognitive abilities like mental rotation and folding are significantly under-tested. Thirdly, the few benchmarks that address dynamic tasks are **insufficient in scale** (Ray et al., 2024; Ramakrishnan et al., 2024), making them insufficient to robustly evaluate the capabilities of the model or to drive further model development.

To bridge these gaps, we propose **Spatial-DISE**. Unlike previous benchmarks that focus on isolated abilities or static scenes, Spatial-DISE adopts a unified 2x2 cognitive taxonomy (Maier, 1996; Uttal et al., 2013), as illustrated in Figure 1 (b), which covers both 2D and 3D aspects, and critically, places a strong emphasis on *dynamic* spatial reasoning tasks. The first dimension distinguishes between **intrinsic** information, which defines an object by its internal parts and their arrangement, and **extrinsic** information, which pertains to the spatial relations among different objects; the second dimension differentiates **static** tasks, which involve fixed and stationary information, from **dynamic** tasks, which require mental transformation. Figure 1 provides an overview of the Spatial-DISE framework, generation pipeline, and benchmark statistics.

**Spatial-DISE** contains more than 12K verified spatial reasoning Visual Question-Answer (VQA) pairs. It is created through a combination of real-world data collection and synthetic generation using Blender[1]. Firstly, it has a set of 559 real-world and synthetic VQA pairs split into 10 different spatial reasoning tasks, covering the four DISE quadrants. Secondly, it includes a set of over 12,000 verified 3D spatial reasoning VQA pairs that are generated through an automated pipeline. The synthetic VQA pairs spread across five 3D Spatial Reasoning tasks.

We conducted a comprehensive evaluation across 32 state-of-the-art (SOTA) VLMs on Spatial-DISE. These encompassed a range of advanced VLMs, featuring both proprietary and open-source models: 22 foundation models, 7 reasoning models, and 3 models post-trained with spatial-related datasets. Our findings reveal a profound and universal weakness in current VLMs. Overall performance remains low, with most models scoring only slightly above random chance and far below the human baseline. Our in-depth error analysis further reveals that these failures stem not from simple visual perception, but from fundamental deficits in cognitive processes like rule-based reasoning and mental simulation. Our key contributions include:

- **A Cognitively Grounded Taxonomy:** We adopt a cognitively grounded framework that, unlike previous task-oriented benchmarks, provides a unified taxonomy to classify any spatial task, revealing specific weaknesses like dynamic reasoning that are otherwise obscured.

- **A Scalable and Verifiable Data Generation Pipeline:** We design and implement a novel, automated pipeline using Blender to programmatically generate complex 3D spatial reasoning tasks. This methodology is a key contribution, offering a reusable tool for the community to overcome the data scarcity that has limited previous dynamic reasoning research. The pipeline ensures verifiability through seeded randomization and reproducible distractor generation strategies.

- **A Unified & Verifiable Cognitive Benchmark:** Leveraging our pipeline, we introduce the Spatial-DISE and the accompanying 12,000-VQA dataset, as a benchmark to systematically

---

[1]https://www.blender.org/

and extensively evaluate complex cognitive spatial reasoning tasks at a scale sufficient for robust evaluation and future model training.

- **Exploring the Boundaries of Cognitive Spatial Reasoning in VLMs:** By benchmarking 32 SOTA models, we define the current boundaries of VLM capabilities in cognitive spatial reasoning. Our analysis reveals a universal performance ceiling, especially for multi-step mental simulation, highlighting the significant gap between AI and human-level spatial intelligence.

## 2 RELATED WORK

**Table 1:** Comparison of Existing Benchmarks under DISE Taxonomy. Abbreviations— I-S: Intrinsic-Static; I-D: Intrinsic-Dynamic; E-S: Extrinsic-Static; E-D: Extrinsic-Dynamic.

| Benchmark | Data Scale | Domain | Source | I-S | I-D | E-S | E-D |
|---|---|---|---|---|---|---|---|
| SpatialRGPT Cheng et al. (2024) | 1k+ | General | Real-World | ✗ | ✗ | ✓ | ✗ |
| BLINK (Fu et al., 2024) | 7k+ | General | Real-World | ✗ | ✗ | ✓ | ✓ |
| VSR (Liu et al., 2023) | 10k | General | Real-World | ✓ | ✗ | ✓ | ✗ |
| What's Up (Kamath et al., 2023) | 820 | General | Real-World | ✗ | ✗ | ✓ | ✗ |
| CV-Bench (Tong et al., 2024) | 2638 | General | Real-World | ✗ | ✗ | ✓ | ✗ |
| LEGO-Puzzles (Tang et al., 2025) | 1100 | Objects | Syn. | ✗ | ✓ | ✓ | ✓ |
| COMFORT (Zhang et al., 2025) | 1220 | Objects | Syn. | ✓ | ✗ | ✓ | ✗ |
| 3DSRBench (Ma et al., 2025) | 2772 | General | Real-World | ✗ | ✗ | ✓ | ✗ |
| VSI-Bench (Yang et al., 2024) | 5k | General | Real-World | ✗ | ✗ | ✗ | ✓ |
| Spatial457 (Wang et al., 2025) | 20k+ | Objects | Syn. | ✓ | ✗ | ✓ | ✓ |
| Q-SpatialBench (Liao et al., 2024) | 271 | General | Real-World | ✗ | ✗ | ✓ | ✗ |
| SAT (Ray et al., 2024) | 175k | General | Real-World+Syn. | ✗ | ✗ | ✓ | ✓ |
| SPARE3D (Han et al., 2020) | 10k+ | Cognition | Syn. | ✓ | ✗ | ✗ | ✗ |
| SpatialEval (Wang et al., 2024) | 13k+ | Cognition | Real-World | ✓ | ✗ | ✓ | ✓ |
| BSA (Xu et al., 2025) | 312 | Cognition | Real-World | ✓ | ✓ | ✓ | ✓ |
| SPACE (Ramakrishnan et al., 2024) | 5k+ | Cognition | Real-World | ✓ | ✓ | ✗ | ✓ |
| OmniSpatial (Jia et al., 2025) | 1.5k | General+Cognition | Real-World | ✓ | ✓ | ✓ | ✓ |
| **Spatial-DISE Bench** | 559 | Cognition | Real-World+Syn. | ✓ | ✓ | ✓ | ✓ |
| **Spatial-DISE-12K** | 12k+ | Cognition | Real-World+Syn. | ✓ | ✓ | ✓ | ✓ |

The evaluation of spatial reasoning ability in VLMs has been an active area of research, but prior work suffers from critical gaps in scope, cognitive depth, and scale. Table 1 compares existing benchmarks in coverage scope, number of instances, and data sources.

Previous benchmarks offer a fragmented evaluation, lacking a unified cognitive framework. Benchmarks such as LEGO-Puzzles (Tang et al., 2025), SAT (Ray et al., 2024) and VSI-Bench (Yang et al., 2024) are confined to narrow, specific tasks, preventing a holistic assessment of a model's true spatial abilities. Spatial-DISE overcomes this by introducing a unified 2x2 cognitive taxonomy. This framework, rooted in cognitive science, enables a comprehensive and balanced evaluation, allowing for the precise diagnosis of model weaknesses.

Furthermore, prior benchmarks have a disproportionate focus on static reasoning. A vast number of benchmarks—including SpatialRGPT (Cheng et al., 2024), SPARE3D (Han et al., 2020), VSR (Liu et al., 2023), CV-Bench (Tong et al., 2024), BLINK (Fu et al., 2024), and What'sUp (Kamath et al., 2023), SpatialEval (Wang et al., 2024) primarily test a model's ability to perceive fixed scenes and relationships. They evaluate what models "see" but not how they can "reason" about potential changes. Spatial-DISE targets this gap by focusing on intricate dynamic reasoning to thoroughly assess cognitive tasks such as 3D rotation and folding.

Finally, while SAT (Ray et al., 2024), SPACE (Ramakrishnan et al., 2024), BSA (Xu et al., 2025) and OmniSpatial (Jia et al., 2025) have begun to explore the dynamic domain, Spatial-DISE's uniqueness lies in its integration of a cognitively unified framework and a verifiable generation process. Our work complements these existing efforts by providing a structured and reproducible approach to understanding model failures. Although one might consider aggregating existing benchmarks into a meta-benchmark, such a union would remain DISE-imbalanced, format-heterogeneous, and unable to provide verifiable training data for the underserved dynamic quadrants—gaps that Spatial-DISE is specifically designed to address.

## 3 METHODOLOGY

Drawing from cognitive science research (Maier, 1996; Uttal et al., 2013), we organize spatial reasoning into two key dimensions: **Intrinsic vs. Extrinsic** and **Static vs. Dynamic**. The first dimension differentiates between **Intrinsic vs. Extrinsic** information. Intrinsic information refers to the essential characteristics and relationships that define an object. Extrinsic information refers to the relation among objects in a group, relative to one another or to an overall framework. The second dimension, **Static vs. Dynamic**, centers on movement. Movement can alter intrinsic information, such as through folding, cutting, or rotation. It can also shift an object's position relative to other objects and the surrounding environment.

This framework comprehensively covers existing task classifications by placing them into four distinct quadrants. This creates a 2x2 taxonomy that categorizes spatial reasoning into four distinct quadrants: **Intrinsic-Static (I-S)** tasks involve analyzing the internal properties of a single, unchanged object; **Extrinsic-Static (E-S)** tasks assess the relationships between multiple objects in a fixed scene; **Intrinsic-Dynamic (I-D)** tasks require mentally simulating transformations on a single object; and **Extrinsic-Dynamic (E-D)** tasks involve reasoning about the changing spatial relationships between multiple objects.

### 3.1 TASKS DESIGN

We designed 10 cognitive science-based tasks to probe spatial reasoning. Figure 2 provides a visual guide to this categorization, showing how various spatial tasks map onto our **Spatial-DISE** taxonomy. The 10 tasks we designed not only fully map to the four quadrants of the DISE framework, but their design inspiration also stems from classical psychometric tests. These tasks are specifically designed to assess core spatial abilities such as mental rotation and spatial visualization (see Appendix A.1 for detailed correspondence).

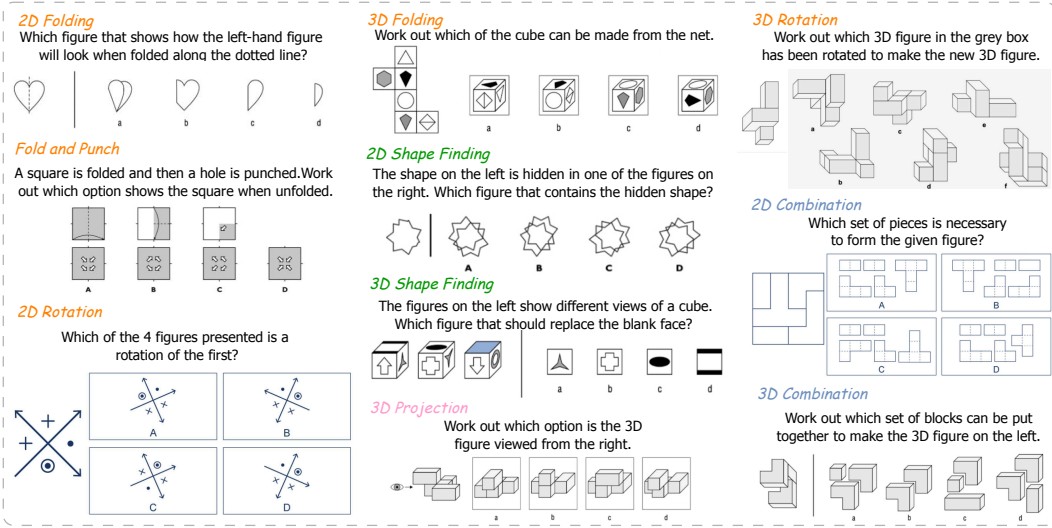

**Figure 2:** 10 Tasks in Spatial-DISE Bench. Orange shows the Intrinsic-Dynamic Tasks, Green shows the Intrinsic-Static Tasks, Pink shows the Extrinsic-Static Tasks and Blue shows the Extrinsic-Dynamic Tasks.

**Intrinsic-Static Tasks.** These tasks evaluate the understanding of an object's fixed, internal spatial properties without transformation. This is assessed through **2D/3D Shape Finding**, which requires identifying a hidden shape within a more complex figure or determining a cube's missing face from other views, thereby testing the static analysis of intrinsic part-whole relationships.

**Intrinsic-Dynamic Tasks.** These tasks test the ability to mentally manipulate the internal properties of an object, requiring pure mental simulation. This includes **2D/3D Rotation**, a classic test of mental transformation that requires predicting an object's appearance after rotation, and **2D/3D Folding & Fold&Punch**, which tests the outcome of folding a 2D net into a 3D shape or unfolding a punched paper.

**Extrinsic-Static Tasks.** These tasks investigate the understanding of fixed spatial relationships from an external viewpoint. This is probed using **3D Projection**, which requires identifying the correct 2D orthographic projection of a 3D object from a specified external direction, a task dependent on the extrinsic relationship between observer and object.

**Extrinsic-Dynamic Tasks.** These tasks assess the ability to reason about the changing relationships between multiple objects or parts. The primary tasks here are **2D/3D Combination**, which require mentally assembling separate parts into a coherent whole and thus tests the ability to simulate how components must move, orient, and connect.

## 3.2 BENCHMARK CURATION

To ensure the scientific rigor and validity of the dataset, a 3-stage curation pipeline was employed. This process integrates real-world data from open sources with a scalable synthetic data generation, followed by human quality control.

**Stage 1: Real-world Data Collection.** The initial phase aimed to establish a conceptual foundation and a repository of templates for subsequent data synthesis. We collected a corpus of existing, high-quality spatial reasoning problems from publicly available and validated sources, including academic psychometric tests and professional aptitude assessments. This phase yielded an initial corpus of 1180 VQA pairs, providing a diverse set of concepts and structures that informed the automated generation process. Detailed real-world data collection is presented in Appendix A.3.

**Stage 2: Scalable Synthetic Data Generation.** As illustrated in Figure 1 d, this core stage was designed to overcome the scale limitations of existing benchmarks, particularly for dynamic and 3D tasks. Leveraging the Blender engine, we transformed the concepts from the initial corpus into a scalable, automated pipeline. This pipeline follows a general paradigm, customized for each of the five 3D task types: **1) Initialization and Seeding:** Each generation task begins with a unique $question\_id$, which is hashed to create a reproducible random seed, ensuring the uniqueness and verifiability of every generated instance. **2) Core Asset Generation:** We generate the core 3D object for a given problem. This includes creating complex, irregular shapes for tasks like 3D Rotation or generating cubes with unique face textures for 3D Folding and 3D Shape Finding. **3) Question and Correct Answer Rendering:** We render the question and correct answer images from optimal camera perspectives. **4) Systematic Distractor Generation**: To ensure the diagnostic challenge of each item, the pipeline implements a suite of tiered strategies to create plausible, near-miss distractors. These strategies include: - *Geometric Variations:* Introducing subtle alterations to the core object's geometry, such as adding or removing components. - *Pattern/Orientation Errors:* Generating incorrect texture layouts or orientations on the faces of an object. - *Incorrect Views:* Rendering a correct object from an incorrect orthographic perspective for projection tasks. - *Component Replacement:* Swapping a correct part with a geometrically similar but incorrect one in assembly tasks. **5) Controlled Rendering and Formatting:** All question, correct answer, and distractors are rendered in a controlled virtual environment with consistent lighting, materials, and camera parameters. The final output is a standardized VQA data pair. Detailed illustration and pseudocode of synthetic data generation is shown in Appendix A.4.

**Stage 3: Rigorous Human Quality Control.** Following generation, all benchmark synthetic instances underwent a rigorous manual verification process to guarantee the benchmark's integrity and reliability. The review protocol assessed each instance against three quality criteria. **1) Solution Uniqueness**: Each problem must have a single, unambiguous correct answer. **2) Accuracy and Clarity**: All images must be free of rendering artifacts, and the corresponding questions must be clearly articulated. All options must be valid according to the task's criteria. **3) Redundancy Elimination**: The instance should not be logically or visually redundant with other items in the dataset. Instances failing to meet these standards were removed from the final dataset. Combined with real-world data, two sets of Spatial-DISE are created:

- **Spatial-DISE Bench**: An evaluation set of 559 carefully selected VQA pairs, covering all 10 task types and four DISE dimensions, designed for model benchmarking.

- **Spatial-DISE 12K**: A large-scale dataset consisting of over 12,000 verifiable VQA pairs cover five 3D tasks, intended as a valuable resource for the future training and fine-tuning of spatial reasoning capabilities in VLMs.

## 4    EVALUATION ON SPATIAL-DISE BENCH

### 4.1    EXPERIMENT SETTING

**Benchmark Models.**    For the evaluation, we select a diverse set of **32** models across **10** model families. Our selection includes both proprietary and open-source models, spanning general foundation models, reasoning models, and spatial-specified models. For proprietary foundation models, we evaluated Claude3.7-Sonnet, Doubao1.5-VL (Guo et al., 2025b), Gemini-2.0-Flash, Gemini-2.5-Flash (w/ and w/o thinking), GPT-4.1-nano, GPT-4o, GPT-4o-mini, GPT-5, and o4-mini. For open-source foundation models, we evaluated InternVL-3-[8B/14B/38B] (Zhu et al., 2025), Llama-3.2-11B-Vision (Grattafiori et al., 2024), Kimi-VL-A3B (Du et al., 2025), Ovis2-[8B/16B] (Lu et al., 2024), Cambrian-[8B/13B] (Tong et al., 2024) and Qwen2.5-VL-[3B/7B/32B] (Bai et al., 2025). For reasoning models, we evaluated LLaVA-CoT (Xu et al., 2024), LMM-R1 (Peng et al., 2025), VLM-R1 (Shen et al., 2025), VLAA-Thinker-[3B/7B] (Chen et al., 2025), Kimi-VL-A3B-Thinking (Du et al., 2025) and Doubao-1.5-thinking (Guo et al., 2025b). For spatial-specified models, we evaluate SpaceThinker (Chen et al., 2024), SpaceOM (Chen et al., 2024) and SpaceR (Ouyang et al., 2025).

**Baseline.**    For comparison, we include Random Guessing and Human Performance. To establish a robust, psychometrically sound human baseline, we recruited 54 diverse participants (ages 15-55) and employed a matrix-sampling design to collect 1,679 valid responses. This design ensured each question was answered by an average of three unique participants. The final human performance is reported as the average accuracy across all responses and cross-validated with Item Response Theory (IRT). More details of human performance in Appendix B.1.

**Implementation Details.**    We evaluate multiple-choice accuracy using exact match via the VLMEvalKit (Duan et al., 2025). Deepseek-R1 (Guo et al., 2025a) is used to parse answers from malformed model outputs. Additional implementation details are provided in the Appendix B.2.

### 4.2    MAIN RESULTS

Our comprehensive evaluation reveals that spatial reasoning remains a significant and universal challenge for current VLMs. Table 2, 3 present the main results of our evaluation. More results are presented in Appendix B.3. We summarize the key findings as followed:

***Spatial reasoning remains a universal challenge.***    The overall performance across all 32 tested models was low, with average accuracy of 28.4%, only marginally above random chance (25%) and falling drastically short of the human baseline (76.8%). Of all models evaluated, the reasoning-enhanced Doubao1.5-VL-thinking achieved the highest overall accuracy at 42.0%. This widespread underperformance indicates a critical weakness in tasks requiring genuine mental transformation, highlighting a failure to move beyond pattern recognition to true spatial cognition.

***Multi-Step transformations overwhelm VLMs reasoning.***    Models demonstrate a particular vulnerability to tasks requiring a sequence of mental transformations. The Fold and Punch task, which requires simulating a fold, a punch, and then an unfold, serves as a clear example of this failure. Even the top-performing model, Doubao-1.5-thinking, only achieved 30.8% accuracy, while the average of all models is only 25.4%, performed near random chance. This indicates that while a model might handle a single transformation, its ability to maintain a coherent mental state breaks down across multiple steps. This suggests a critical deficit in "spatial working memory," preventing models from reliably tracking an object through a sequence of changes.

***Post-training shows improvement but not enough.***    The results reveal that post-training with reinforcement learning or fine-tuning on spatial datasets offers limited improvements. While models like Doubao-1.5-thinking and SpaceThinker showed performance gains, their absolute accuracies remain low and far from the human baseline.

***Static comprehension is not a solved precursor to dynamic reasoning.***    Counter-intuitively, the results show that proficiency in static reasoning is not a prerequisite for dynamic reasoning. Several top models perform better on dynamic tasks than static ones. For example, Gemini2.0-Flash scored significantly higher on dynamic tasks (38.3%) than on static tasks (23.6%). Doubao-1.5-thinking even outperform human performance in Extrinsic-Dynamic questions. This suggests that models are

**Table 2:** Evaluation results of 32 SOTA models and 2 models SFT on Spatial-DISE. Row colors: Base, Δ vs base, Reasoning, Spatial, SFT on Spatial-DISE-12k. A Δ row shows the *absolute change in percentage points (pp)* relative to its base model and is placed between the parent and the derived model. Values are accuracy (%); brackets use [lower, upper] for the 95% CI. **Bold** indicates the highest accuracy; Underline indicates the second highest.

| Model Tree | | Acc. [95% CI] | E-D [95% CI] | E-S [95% CI] | I-D [95% CI] | I-S [95% CI] |
|---|---|---|---|---|---|---|
| *Proprietary Bases* | | | | | | |
| **Claude 3.7 Sonnet** | *Base* | 30.6% [26.8, 34.3] | 22.6% [14.3, 32.1] | 31.4% [21.4, 42.9] | 32.4% [27.4, 37.7] | 31.0% [21.8, 41.4] |
| **Doubao1.5VL** | *Base* | 33.8% [29.9, 37.7] | 31.0% [21.4, 40.5] | 37.1% [25.7, 48.6] | 33.6% [28.6, 39.0] | 34.5% [25.3, 44.8] |
| \| | | (↑8.2) | (↑30.9) | (↓5.7) | (↑7.3) | (↑1.1) |
| \|− Doubao1.5VL-thinking | RLHF+RLVF | 42.0% [37.9, 46.2] | 61.9% [51.2, 72.6] | 31.4% [21.4, 42.9] | 40.9% [35.5, 46.2] | 35.6% [25.3, 46.0] |
| **Gemini 2.0 Flash** | *Base* | 34.2% [30.4, 37.9] | 25.0% [15.5, 34.5] | 31.4% [21.4, 42.9] | 41.8% [36.5, 47.2] | 17.2% [9.2, 25.3] |
| **Gemini 2.5 Flash** | *Base* | 31.5% [27.7, 35.2] | 16.7% [9.5, 25.0] | 27.1% [17.1, 37.1] | 39.3% [33.9, 44.7] | 20.7% [12.6, 29.9] |
| **Gemini 2.5 Flash w/o thinking** | *Base* | 32.0% [28.3, 35.8] | 15.5% [8.3, 23.8] | 28.6% [18.6, 38.6] | 39.6% [34.3, 45.0] | 23.0% [14.9, 32.2] |
| **GPT-4.1 nano** | *Base* | 29.3% [25.6, 33.1] | 29.8% [20.2, 40.5] | 35.7% [25.7, 47.1] | 31.1% [26.1, 36.2] | 17.2% [9.2, 25.3] |
| **GPT-4o** | *Base* | 28.1% [24.5, 31.8] | 26.2% [16.7, 35.7] | 22.9% [12.9, 32.9] | 29.9% [24.8, 34.9] | 27.6% [18.4, 36.8] |
| **GPT-4o-mini** | *Base* | 25.6% [22.0, 29.2] | 16.7% [9.5, 25.0] | 21.4% [12.8, 31.4] | 28.0% [23.0, 33.0] | 28.7% [19.5, 37.9] |
| **GPT-5** | *Base* | 30.1% [26.3, 34.0] | 23.8% [15.5, 33.3] | 25.7% [15.7, 35.7] | 33.6% [28.6, 39.0] | 26.4% [17.2, 35.6] |
| **o4-mini** | *Base* | 33.3% [29.5, 37.2] | 16.7% [9.5, 25.0] | 25.7% [15.7, 35.7] | 36.8% [31.8, 42.1] | 42.5% [32.2, 52.9] |
| *Proprietary Average* | | *31.9% [29.0, 34.7]* | *26.0% [17.2, 34.8]* | *28.9% [25.6, 32.3]* | *35.2% [32.0, 38.4]* | *27.7% [22.3, 33.0]* |
| *Open-source Bases* | | | | | | |
| **Llama3V-11B** | *Base* | 24.5% [20.9, 28.1] | 29.8% [20.2, 39.3] | 14.3% [7.1, 22.9] | 25.5% [20.8, 30.5] | 24.1% [14.9, 33.3] |
| \| | | (↓0.5) | (-) | (↑8.6) | (↓1.0) | (↓6.9) |
| \|− LLaVA-CoT | CoT | 24.0% [20.6, 27.5] | 29.8% [20.2, 39.3] | 22.9% [12.9, 32.9] | 24.5% [19.8, 29.2] | 17.2% [10.3, 25.3] |
| **Cambrian-13B** | *Base* | 26.7% [23.1, 30.4] | 25.0% [16.7, 34.5] | 32.9% [21.4, 44.3] | 25.8% [21.1, 30.8] | 26.4% [17.2, 35.6] |
| **Cambrian-8B** | *Base* | 22.9% [19.5, 26.3] | 19.0% [10.7, 27.4] | 15.7% [7.1, 24.3] | 23.9% [19.2, 28.6] | 28.7% [19.5, 37.9] |
| **InternVL3-38B** | *Base* | 32.4% [28.6, 36.3] | 27.4% [17.9, 36.9] | 30.0% [20.0, 41.4] | 35.8% [30.8, 41.2] | 26.4% [17.2, 35.6] |
| **InternVL3-14B** | *Base* | 31.1% [27.4, 34.9] | 21.4% [13.1, 29.8] | 31.4% [20.0, 42.9] | 37.1% [31.8, 42.5] | 18.4% [10.3, 26.4] |
| **InternVL3-8B** | *Base* | 26.3% [22.7, 29.9] | 23.8% [15.5, 33.3] | 28.6% [18.6, 38.6] | 30.8% [25.8, 35.8] | 10.3% [4.6, 17.2] |
| **Kimi-VL-A3B** | *Base* | 24.3% [20.8, 27.9] | 17.9% [9.5, 26.2] | 27.1% [17.1, 37.1] | 27.7% [22.6, 32.7] | 16.1% [9.2, 24.1] |
| \| | | (↑0.4) | (↑9.5) | (↑1.5) | (↓3.8) | (↑5.7) |
| \|− Kimi-VL-Thinking | CoT+RL | 24.7% [21.1, 28.3] | 27.4% [17.9, 36.9] | 28.6% [18.6, 38.6] | 23.9% [19.2, 28.6] | 21.8% [13.8, 31.0] |
| **Ovis2-16B** | *Base* | 26.3% [22.7, 29.9] | 20.2% [11.9, 28.6] | 27.1% [17.1, 38.6] | 31.4% [26.4, 36.8] | 12.6% [5.7, 19.5] |
| **Ovis2-8B** | *Base* | 23.8% [20.4, 27.4] | 15.5% [8.3, 23.8] | 21.4% [12.9, 31.4] | 29.6% [24.5, 34.6] | 12.6% [5.7, 20.7] |
| **Qwen2.5-VL-32B** | *Base* | 27.2% [23.4, 30.9] | 21.4% [13.1, 29.8] | 31.4% [21.4, 42.9] | 27.7% [23.0, 32.7] | 27.6% [18.4, 37.9] |
| **Qwen2.5-VL-7B** | *Base* | 26.1% [22.5, 29.9] | 32.1% [22.6, 42.9] | 24.3% [14.3, 34.3] | 27.7% [22.6, 32.7] | 16.1% [9.2, 24.1] |
| \| | | (↑1.8) | (↓4.7) | (↑2.8) | (↑0.9) | (↑10.3) |
| \|− VLAA-Thinker-7B | GRPO | 27.9% [24.3, 31.7] | 27.4% [17.9, 36.9] | 27.1% [17.1, 37.1] | 28.6% [23.9, 33.6] | 26.4% [17.2, 35.6] |
| \| | | (↑20.9) | (↑34.6) | (↑11.4) | (↑15.4) | (↑35.6) |
| \|− Qwen2.5-VL-7B-sft | SFT (SD-12k) | **47.0%** [42.9, 51.2] | **66.7%** [56.0, 76.2] | **35.7%** [24.3, 47.1] | **43.1%** [37.7, 48.7] | 51.7% [41.4, 62.1] |
| \| | | (↑0.9) | (↓2.3) | (↓7.2) | (↑1.9) | (↑6.9) |
| \|− SpaceR | SG-RLVR | 27.0% [23.4, 30.8] | 29.8% [20.2, 39.3] | 17.1% [8.6, 27.1] | 29.6% [24.5, 34.6] | 23.0% [14.9, 32.2] |
| **Qwen2.5-VL-3B** | *Base* | 22.9% [19.5, 26.5] | 25.0% [15.5, 34.5] | 17.1% [8.6, 25.7] | 26.4% [21.7, 31.4] | 12.6% [5.7, 20.7] |
| \| | | (↑3.2) | (↑4.8) | (↑2.9) | (-) | (↑13.8) |
| \|− LMM-R1 | PPO | 26.1% [22.5, 29.9] | 29.8% [20.2, 39.3] | 20.0% [11.4, 30.0] | 26.4% [21.7, 31.4] | 26.4% [17.2, 35.6] |
| \| | | (↑7.9) | (↑8.3) | (↑1.5) | (↑7.2) | (↑15.0) |
| \|− VLM-R1 | GRPO | 30.8% [27.0, 34.7] | 33.3% [23.8, 44.0] | 18.6% [10.0, 28.6] | 33.6% [28.6, 39.0] | 27.6% [18.4, 36.8] |
| \| | | (↑3.0) | (↑3.6) | (↑12.9) | (↑1.3) | (↑1.2) |
| \|− VLAA-Thinker-3B | GRPO | 25.9% [22.4, 29.5] | 28.6% [19.0, 38.1] | 30.0% [20.0, 41.4] | 27.7% [23.0, 32.7] | 13.8% [6.9, 21.8] |
| \| | | (↑6.7) | (↑10.7) | (↑7.1) | (↑10.7) | (↑11.5) |
| \|− SpaceThinker | SFT | 32.6% [25.4, 32.9] | 39.3% [20.2, 40.5] | 22.9% [15.7, 35.7] | 34.9% [27.7, 37.7] | 25.3% [10.3, 26.4] |
| \| | | - | (↑2.4) | (↓5.7) | (↓1.0) | (↑5.7) |
| \|− SpaceOM | SFT | 25.9% [22.4, 29.5] | 31.0% [20.2, 39.3] | 24.3% [14.3, 34.3] | 26.7% [22.0, 31.8] | 19.5% [11.5, 28.7] |
| \| | | (↑15.4) | (↑21.4) | (↑2.8) | (↑11.0) | (↑35.7) |
| \|− SpaceOM-sft | SFT (SD-12k) | 41.3% [37.4, 45.4] | 52.4% [41.7, 63.1] | 27.1% [17.1, 37.1] | 37.7% [32.4, 43.1] | **55.2%** [44.8, 65.5] |
| *Open-source Average* | | *26.2% [25.2, 27.3]* | *23.2% [22.4, 27.3]* | *25.1% [22.2, 28.0]* | *29.1% [27.7, 30.6]* | *19.3% [17.0, 21.7]* |
| **Human Level** | | 76.8% [74.8, 78.9] | 61.1% [56.6, 65.5] | 81.1% [76.7, 85.5] | 80.2% [78.2, 82.3] | 76.8% [72.9, 80.7] |
| **Random Guessing** | | 24.8% | 25.4% | 26.3% | 24.3% | 24.7% |

not learning spatial reasoning in a human-like, scaffolded manner. Instead of building dynamic capabilities upon a solid foundation of static scene understanding, they appear to be learning fragmented strategies, recognizing patterns of "change" without a robust, underlying model of the static world.

***Computational rigor can outperform fallible human simulation.*** This is evident in Doubao-1.5-thinking model, which surpassed the human baseline on E-D tasks. This superior performance can likely be attributed to the nature of 2D/3D combination. As confirmed by our human performance analysis (Appendix, Table 9), these tasks are particularly arduous and cognitively demanding for humans. 3D Combination commanding the longest mean response time (59.2s) among all tasks for humans, who must rely on fallible mental simulation. In contrast, we observed that the Doubao model transforms these challenges into computational problems by employing a more algorithmic strategy to compute and compare geometric features of components—such as edges, angles, and connection points. Essentially, the model excels by converting a cognitively exhausting simulation task into a precise computational problem, a domain where it holds a distinct advantage over human intuition.

## 4.3 FINE-TUNING ON SPATIAL-DISE-12K

To ascertain whether Spatial-DISE-12K effectively enhances the spatial reasoning capabilities of vision-language models, and to evaluate the cross-domain generalizability of this training, we conducted supervised fine-tuning experiments. Specifically, two representative open-source VLMs—Qwen2.5-VL-7B and SpaceOm—were fine-tuned using Low-Rank Adaptation (LoRA) applied to all linear layers utilizing the Spatial-DISE-12K training split (see Appendix B.4 for implementation details). Following optimization, we structure our comprehensive evaluation around four core research questions (RQs), assessing the models on the Spatial-DISE Bench alongside five established external benchmarks: CVBench, SAT, SPACE, OmniSpatial, and VSIBench_MCQ.

**Table 3:** Accuracy for Qwen2.5-VL (Base vs SFT) and SpaceOm (Base vs SFT). Δ is SFT-Base in percentage points (pp).

|  | Spatial-DISE | CVBench | SAT | SPACE | OmniSpatial | VSIBench_MCQ |
|---|---|---|---|---|---|---|
| SpaceOm | 25.9% | 68.8% | 46.67% | 27.22% | 27.91% | 31.05% |
| +DISE SFT | 41.3% | 70.33% | 49.33% | 32.6% | 34.28% | 33.7% |
| Δ | (↑15.4%) | (↑1.53%) | (↑2.66%) | (↑5.38%) | (↑6.37%) | (↑2.65%) |
| Qwen2.5-VL-7B | 26.1% | 75.9% | 65.3% | 28.7% | 21.8% | 19.3% |
| +DISE SFT | 47.0% | 77.4% | 69.3% | 32.2% | 34.0% | 22.6% |
| Δ | (↑20.9%) | (↑1.5%) | (↑4.0%) | (↑3.5%) | (↑12.2%) | (↑3.3%) |

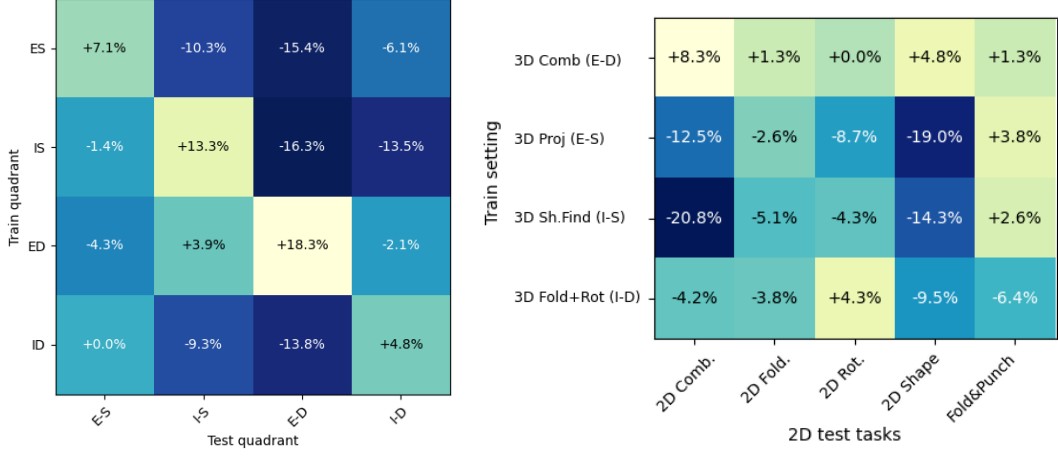

(a) Cross-quadrant transfer.    (b) 3D→2D transfer across DISE tasks.

**Figure 3:** Transfer heatmap by fine-tuning Qwen2.5-VL-7B on Spatial-DISE quadrant-wise.

***RQ1: Does 3D spatial training cultivate a broad, structured set of in-domain skills?*** To address this, we evaluated the fine-tuned models on the Spatial-DISE Bench. Empirical evaluation demonstrates that fine-tuning on Spatial-DISE-12K yields substantial performance gains. Specifically, the overall accuracy of Qwen2.5-VL-7B improved from 26.1% to 47.0%, and SpaceOm from 25.9% to 41.3%. The most pronounced enhancements were observed in Intrinsic-Dynamic and Extrinsic-Dynamic tasks, alongside a remarkable surge in Qwen2.5-VL-7B's Intrinsic-Static performance (scaling from 16.1% to 51.7%).

***RQ2: Do models learn factorized spatial dimensions or entangled, quadrant-specific representations?*** To elucidate the underlying mechanisms driving these gains, we isolated the training curriculum by fine-tuning Qwen2.5-VL-7B exclusively on individual DISE quadrants and visualized the subsequent accuracy fluctuations (Figure 3a). The resulting heatmap reveals a robust diagonal pattern, indicating profound quadrant-specific specialization: training on a specific quadrant predominantly amplifies performance within that corresponding domain. Conversely, off-diagonal transfer is largely minimal or even detrimental, failing to exhibit clean factorization along the Intrinsic/Extrinsic or Static/Dynamic axes. We observe distinct directional asymmetries (e.g., a mild positive transfer of +3.9 pp from 3D E-D to I-S, which starkly contrasts with the strongly negative transfer from I-S to E-D). This implies that the four DISE quadrants encapsulate relatively distinct cohorts of spatial

competencies; strengthening one family does not automatically confer a "universal" spatial skill, and overall benchmark gains necessitate exposure to multiple 3D quadrants during training.

***RQ3: Can 3D spatial training induce representations that transfer to 2D reasoning tasks?*** To probe cross-dimensional transferability, we maintained the single-quadrant fine-tuning paradigm but evaluated the resultant models across all 2D DISE tasks (Figure 3b). The findings expose qualitatively divergent reasoning regimes. Training on extrinsic-dynamic 3D tasks catalyzed broadly positive transfer across varied 2D settings, suggesting that scene-centric dynamic reasoning fosters highly reusable representations applicable to 2D projection, combination, and occlusion problems. In stark contrast, training on intrinsic-static or extrinsic-static 3D tasks yielded narrowly constrained or even negative transfer. Furthermore, while intrinsic-dynamic training facilitated 2D rotation performance, it concurrently degraded the model's capacity to solve simpler 2D problems. These asymmetric patterns underscore that object-centric static reasoning tends to form overly specialized schemas, which can actively interfere with tasks reliant on relative or dynamic reference frames.

***RQ4: Do the acquired spatial competencies generalize to external benchmarks without catastrophic forgetting?*** To assess broader applicability, we evaluated the fine-tuned models on a suite of external spatial benchmarks (Table 3). Fine-tuning on Spatial-DISE engendered consistent yet highly selective out-of-domain gains, importantly demonstrating that this targeted spatial curriculum does not induce catastrophic forgetting of general visual capabilities (e.g., Qwen2.5-VL-7B-sft improved marginally on CVBench). The most substantial improvements materialized on the SPACE and OmniSpatial benchmarks, which intrinsically emphasize viewpoint transformations and 3D-consistent reasoning. Conversely, benchmarks amalgamating spatial reasoning with generalized language or diagrammatic comprehension exhibited more modest benefits. This corroborates our prior analyses: Spatial-DISE-12K fortifies specific spatial reasoning architectures that subsequently transfer primarily to structurally homologous external tasks.

Ultimately, even following rigorous fine-tuning, the apex model (Qwen2.5-VL-7B-sft) remains markedly inferior to the human baseline on the Spatial-DISE Bench, highlighting substantial avenues for future architectural refinement. Collectively, the responses to these research questions demonstrate that while contemporary VLMs do not yet possess robust, human-like spatial schemas, meticulously structured 3D training regimens like Spatial-DISE-12K can systematically cultivate distinct spatial skill cohorts and induce meaningful, albeit selective, cross-task and cross-benchmark transfer.

## 5 ERROR ANALYSIS

### 5.1 ERROR TAXONOMY AND MAIN DEFICITS

Given the persistent performance gap between fine-tuned VLMs and human baselines , we must move beyond simply measuring what models fail at to understanding why they fail. To provide this cognitive diagnosis, we employ Doubao-1.6-thinking as an automated judge and, combined with human analysis, analyze a sample of 200 incorrect responses from four representative models: **GeminiFlash2-0**, **Qwen2.5-VL-3B**, **Doubao-1.5-thinking**, and **SpaceThinker**, with 50 samples drawn from each.

We established a high-level error taxonomy to systematically diagnose failures by deconstructing the model mistakes into three errors: **Perceptual Error**, **Comprehension Error** and **Reasoning Error**. The analysis reveals that Reasoning errors are the predominant failure category, accounting for an overwhelming 72.5% of all analyzed failure responses. Perceptual errors constituted 17.5% of the total, while comprehension errors were the least common at 10%. This distribution strongly suggests that the primary bottleneck for current VLMs is not in visual perception but in complex spatial-logical inference. The predominance of reasoning errors (145) prompted a deeper analysis, which identified three fundamental cognitive deficits.

**Table 4:** Error Types and Their Frequencies.

| Major Error | Sub-category | Num. |
| --- | --- | --- |
| Reasoning Err. | Failure in Rule Application | 65 |
| | Failure in Mental Simulation | 58 |
| | Failure in Holistic-Local Processing | 22 |
| Perceptual Err. | — | 35 |
| Comprehension Err. | — | 20 |

The most significant issue was a **Failure in Rule Application** (44.8%), where models disregard basic geometric axioms, such as the spatial relationship between adjacent and opposite faces on a cube. This suggests an inability to link visual data with abstract principles. The second major deficit was a **Failure in Mental Simulation** (40.0%), indicating a lack of "spatial working memory" to track objects through transformations, as seen in Fold and Punch where state changes are consistently miscalculated. Finally, a **Failure in Holistic-Local Processing** (15.2%) was observed, where models cannot appropriately shift attention between an object's overall structure and its local details, often being misled by superficial similarities while ignoring critical flaws. Detailed error analysis pipeline, definition of error categories and more discussion is presented in Appendix C.

## 5.2 DISENTANGLING PERCEPTUAL CONFOUNDERS

**Table 5:** Layout Ablation Study on Qwen2.5-VL-7B to Disentangle Perceptual Confounders.

| Input Setting | Accuracy |
|---|---|
| Standard Merged Layout | 26.1% |
| Separate Images Layout | 24.9% |

Although the preceding analysis categorizes the overwhelming majority of failures as reasoning deficits, confirming the absolute rigor of this conclusion requires eliminating potential perceptual bottlenecks. To ensure that our error analysis accurately captures genuine spatial reasoning deficits rather than superficial perceptual confounders—such as an inability to parse complex, multi-panel visual layouts—we conducted a layout ablation study. In our standard evaluation setting, the question and corresponding options are presented to the model as a single merged image. It is plausible that a model's failure could stem from visual parsing issues rather than spatial reasoning limits.

To test this, we compared the standard merged-image setting against a separate-images setting, where the question and each option are provided to the model as independent image inputs. Evaluating Qwen2.5-VL-7B under these two conditions yielded near-identical accuracies: 26.1% for the standard merged layout and 24.9% for the separate images. If layout parsing were the primary bottleneck, isolating the images should have resulted in a clear performance improvement. The absence of such a gain strongly supports our qualitative findings that the dominant difficulty for VLMs lies in complex spatial-logical inference—such as maintaining 3D adjacencies and global consistency—rather than format parsing.

## 6 CONCLUSION AND LIMITATIONS

**Conclusion and future direction.** We introduced **Spatial-DISE** and Spatial-DISE-12K to comprehensively evaluate spatial reasoning in VLMs. Our findings reveal universal cognitive deficits, particularly in applying geometric rules and performing mental simulations. This work provides a necessary framework to guide future VLM development.

To advance spatial intelligence, future research must shift from mere visual perception to active, human-like reasoning. Key directions include: **(1)** closing the sim-to-real gap by transferring abstract cognitive concepts derived from synthetic settings; **(2)** evolving evaluations from isolated puzzles to interactive tasks (e.g., navigation and robot manipulation) ; and **(3)** adopting process-oriented assessments that require textual justifications or action plans, distinguishing true mental simulation from fragile heuristic matching

**Limitations.** Our error analysis relies on a hybrid LLM+human pipeline in which Doubao-1.6-thinking proposes an error type and explanation that a human annotator then verifies. While this reduces manual effort, the LLM's initial label may bias the annotator, and analysing only 200 errors from four models means Table 4 should be read as qualitative trends rather than precise estimates. Additionally, Spatial-DISE-12K consists entirely of synthetic 3D scenes; despite consistent gains on real-world benchmarks, a sim-to-real gap may persist for naturalistic settings. Human quality control covers a stratified sample of 1,000 items, with programmatic single-solution checks applied to the remainder, as exhaustive manual verification of the full 12K is infeasible.

## ACKNOWLEDGMENTS

**LLM Usage Statement.** We declare that large language models (LLMs) were used exclusively for language editing and stylistic improvements in this manuscript. They did not contribute to the conceptual, methodological, or experimental aspects of the work.

**Ethics Statement.** This work adheres to ethical research practices. Human performance data were collected from 54 participants. The study protocol was reviewed and approved by the Faculty of Science and Engineering Research Ethics Committee at the University of Liverpool (Project ID: 16623). Prior to participation, all individuals were informed of the study's purpose and procedure. They provided digital informed consent before accessing the tasks and were explicitly notified of their right to withdraw at any time. No sensitive personal data was collected. All response data are anonymized and will be destroyed three months after the paper is accepted.

**Reproducibility Statement.** We provide comprehensive details to ensure full reproducibility. The complete dataset curation process, including the synthetic data generation pipeline, is detailed in Section 3.2 and Appendix A.4. This includes procedural algorithms (pseudocode) and specific implementation details for the five core 3D tasks. All evaluation settings, including benchmark models, baselines, and implementation details, are described in Section 4.1. Our human performance assessment methodology is thoroughly documented in Appendix B.1. The benchmark, dataset, and code will be made publicly available to facilitate direct comparison with our results.

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

# Appendices

## A  DATASET DETAILS

### A.1  TASKS DESIGN DETAILS

This subsection describes the task design details, aligning the original cognitive science psychometric test with the spatial abilities defined by Linn & Petersen (1985), and its classification within the Spatial-DISE taxonomy.

It is important to note that the Spatial-DISE taxonomy is intentionally a high-level, problem-centric abstraction. It categorizes spatial reasoning tasks based on their core structural properties and primary objectives—specifically, whether the problem relies on intrinsic versus extrinsic information, and whether the goal state requires static analysis or dynamic transformation.

However, human solvers and models may employ a variety of lower-level, atomic cognitive strategies to solve these tasks. For instance, while "3D Shape Finding" is structurally classified as an Intrinsic-Static task (as the objective is to determine a fixed intrinsic property of a single object), a viable solution strategy might involve mentally rotating the object (an Intrinsic-Dynamic process) to build a complete mental model. The DISE categorization reflects the dominant structural requirement of the task itself, whereas the classic spatial abilities (such as Mental Rotation [MR] or Spatial Visualization [SV] detailed in Table 5) represent the atomic skills that can be invoked across multiple quadrants. Therefore, our evaluation uses DISE to provide a coarse-grained organizational framework over task families, diagnosing systematic weaknesses along structural dimensions rather than isolating a single low-level cognitive process.

**Table 6:** Each spatial task used in our study and its canonical source test. Spatial Perception (SP), Spatial Relation (SR), Spatial Orientation (SO), Mental Rotation (MR), and Spatial Visualization (SV)

| Task | Original Test | DISE Taxonomy | Spatial Ability |
|---|---|---|---|
| 3D Combination | Differential Aptitude Tests (Bennett et al., 1947) | Extrinsic-Dynamic | SV |
| 2D Combination | Minnesota Paper Form Board Test (Peter, 1974) | Extrinsic-Dynamic | SV |
| 3D Projection | Purdue Spatial Visualization Test – Views (BODNER & GUAY, 1997) | Extrinsic-Static | SP, SV |
| Fold and Punch | Paper Folding Test (VZ-2) (Pallrand & Seeber, 1984; Ekstrom & Harman, 1976) | Intrinsic-Dynamic | SV, SR |
| 3D Folding | Paper Folding Test (VZ-3) (Ekstrom & Harman, 1976) | Intrinsic-Dynamic | SV, SR, SO |
| 2D Folding | Paper Folding Test (VZ-2) (Ekstrom & Harman, 1976) | Intrinsic-Dynamic | SV, SR, SO |
| 3D Rotation | Mental Rotations Test (Shepard & Metzler, 1971) | Intrinsic-Dynamic | SV, MR, SO |
| 2D Rotation | Card Rotations Test (S-1) (Ekstrom & Harman, 1976) | Intrinsic-Dynamic | SV, MR, SO |
| 3D Shape Finding | Cube Comparisons Test (Ekstrom & Harman, 1976) | Intrinsic-Static | SV, SR |
| 2D Shape Finding | Embedded Figures Test (Witkin, 1950) | Intrinsic-Static | SV, SR |

### A.2  DATASET SPLIT DETAILS

| Subset | Q&A Pairs | Source Mix (RWD / SD) | Tasks |
|---|---|---|---|
| Spatial-DISE-Bench | 559 | 53% / 47% | 2D + 3D |
| Spatial-DISE-12K | 12355 | 5% /95% | 3D |
| -Train | 8648 | 5.1% / 94.9% | 3D |
| -Val | 1853 | 5.5% / 94.5% | 3D |
| -Test | 1854 | 4.5% / 95.5% | 3D |

**Table 7:** Description of Spatial-DISE Subsets. RWD: Real-World Data, SD: Synthetic Data. Note that Spatial-DISE Bench includes 2D questions absent from training splits, enabling zero-shot 2D evaluation.

### A.3  REAL-WORLD DATA COLLECTION DETAILS

Real-world data are collected from open source online resources, mainly from the following resources:

1. Open-Source Spatial Reasoning Tests: Psychometric test materials published by academic research entities for evaluating spatial abilities.

2. CEM 11+ Non-verbal Reasoning Tests: Validated spatial reasoning items from authoritative aptitude tests used for secondary school admissions in the UK.

3. Online Employment Aptitude Tests: High-quality spatial and logical problems administered by corporations during recruitment.

## A.4 Synthetic Data Generation Details

**Figure 4:** Synthetic Data Generation and Quality Control.

This section provides detailed procedural algorithms (pseudocode) and visual examples for the automated generation of our five core 3D spatial reasoning tasks. Each algorithm is designed for verifiability and incorporates sophisticated, task-specific strategies for generating plausible distractors.

Synthetic data generation employs Blender 4.4.0 on Apple Silicon M4. Some texture icons © Icons8 — under Universal Multimedia License. Task details, pseudocode[2], and examples for synthetic data generation are outlined below:

---

**Algorithm 1: GENERATE3DROTATIONQUESTION**

1: **Input:** question id q_id, list of isometric camera presets $iso\_views$, number of distractors $n$
2: $seed \leftarrow$ Hash(q_id)
3: SetRandomSeed($seed$); ClearScene()
4: $orig \leftarrow$ CreateCombinationShape($cells \in [5, 15]$, $rectangularPrisms$=True, $seed$)
5: $qView \leftarrow$ FindBestView($orig$, $iso\_views$)
6: SetCamera($qView$, $jitter$=True)
7: RenderImage(q_id_$Q$)
8: $ansView \leftarrow$ ChooseDifferentView($iso\_views$, exclude=$qView$)
9: SetCamera($ansView$, $jitter$=True)
10: RenderImage(q_id_$A0$)
11: **for** $i \leftarrow 1$ **to** $n$ **do**
12: $difficulty \leftarrow i/n$ {Higher $i \Rightarrow$ harder}
13: $dShape \leftarrow$ GenerateDistractor($orig$, $difficulty$, $seed + i$)
14: $dView \leftarrow$ RandomChoice($iso\_views$)
15: SetCamera($dView$, $jitter$=True)
16: RenderImage(q_id_$A\{i\}$)
17: **end for**
18: SaveMetadata({q_id, $qView$, $ansView$, $seed$})

---

[2]All functions referenced in the code listings are project-specific utility routines; their full implementations will be provided in the accompanying public code repository.

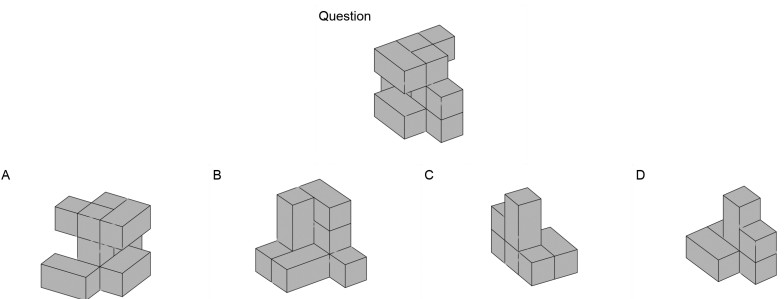

**Figure 5:** Synthetic 3D Rotation Data Example.

**3D Rotation**    The 3D rotation matching task is designed to assess the ability to mentally rotate a three-dimensional object and recognize it from a different angle.

The process begins by generating a complex 3D shape composed of multiple cubes or rectangular prisms. This shape is then rendered from an optimal viewpoint to create the "question" image. This viewpoint is chosen to maximize the number of visible parts, ensuring a clear presentation of the object.

Next, a set of "answer" options is generated:

The Correct Answer: This is created by rendering the original shape from a new viewpoint, different from the one used for the question image. This requires the participant to recognize that it is the same object, despite the change in perspective.

Distractors: These are generated by creating new shapes that are slightly different from the original one. Each distractor is then rendered from a different viewpoint. These are designed to confuse the participant by presenting options that are visually similar but structurally incorrect. The final output consists of the question image, one correct answer image, and several distractor images, along with a metadata file containing all the generation parameters to ensure reproducibility.

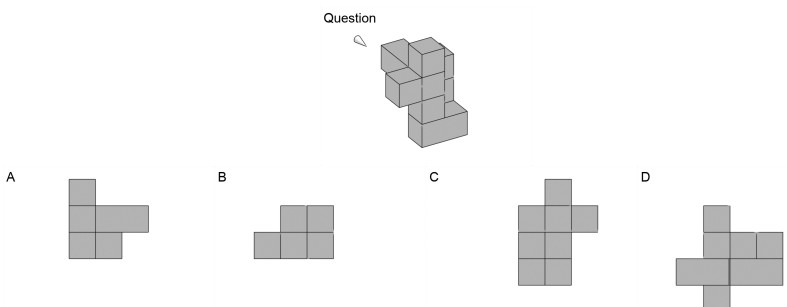

**Figure 6:** Synthetic 3D Projection Data Example.

**3D Projection**    The 3D projection task evaluates a person's ability to interpret a 3D object from an isometric perspective and then identify its correct 2D orthographic projection from a set of options.

The process starts by generating a complex 3D shape. A "question" image is then created by rendering this 3D shape from an optimal isometric viewpoint. A visual cue, typically an arrow, is included in the question image to indicate the direction from which the orthographic projection should be imagined (e.g., "top-down," "front," or "side" view).

A set of options is then generated:

The Correct Answer: This is the true 2D orthographic projection of the 3D shape as seen from the direction indicated by the arrow in the question image.

Distractors: These are incorrect 2D projections. They are generated in a few ways:

---

**Algorithm 2:** GENERATE3DPROJECTIONQUESTION

---

1: **Input:** q_id, $ortho\_views = \{top, front, right, left, bottom, back\}$, $iso\_views$, number of distractors $n$
2: $seed \leftarrow$ Hash(q_id)
3: SetRandomSeed($seed$); ClearScene()
4: $shape \leftarrow$ CreateCombinationShape(seed=$seed$)
5: $qView \leftarrow$ FindBestView($shape$, $iso\_views$)
6: $targetView \leftarrow$ RandomChoice($ortho\_views$)
7: $indicator \leftarrow$ CreateViewIndicator(direction=$targetView$)
8: SetCamera($qView$)
9: RenderImage(q_id_Q); Delete($indicator$)
10: SetCameraOrtho($targetView$)
11: RenderImage(q_id_A0)
12: **for** $i \leftarrow 1$ **to** $n$ **do**
13:     **if** Random() < 0.7 **then**
14:         $dShape \leftarrow$ GenerateDistractor($shape$, difficulty=$0.3 + 0.7 \cdot i/n$, seed+$i$)
15:         $dView \leftarrow targetView$
16:     **else**
17:         $dShape \leftarrow shape$
18:         $dView \leftarrow$ ChooseDifferentView($ortho\_views$, exclude=$targetView$)
19:     **end if**
20:     ApplyScene($dShape$)
21:     SetCameraOrtho($dView$)
22:     RenderImage(q_id_A$\{i\}$)
23: **end for**
24: SaveMetadata($\{$q_id, $seed$, $qView$, $targetView\}$)

---

Incorrect Projections: These are valid orthographic projections but from the wrong viewpoint (e.g., a "side" view when the "top-down" view was asked for).

Slightly Altered Shapes: These are 2D projections of shapes that are subtly different from the original 3D shape, testing attention to detail. The participant must select the 2D image that accurately represents the specified orthographic projection of the 3D object shown in the question.

---

**Algorithm 3:** GENERATE3DCOMBINATIONQUESTION

---

1: **Input:** q_id, $iso\_views$, number of distractors $n$
2: $seed \leftarrow$ Hash(q_id)
3: SetRandomSeed($seed$); ClearScene()
4: $master \leftarrow$ CreateCombinationShape(seed=$seed$, complexity=medium)
5: $qView \leftarrow$ FindBestView($master$, $iso\_views$); SetCamera($qView$)
6: RenderImage(q_id_Q)
7: $oppView \leftarrow$ OppositeView($qView$); SetCamera($oppView$)
8: RenderImage(q_id_Q_opp)
9: $components \leftarrow$ DeconstructShape($master$)
10: ArrangeComponentsGrid($components$, gap=2)
11: SetCamera(GlobalOverview)
12: RenderImage(q_id_A0)
13: **for** $i \leftarrow 1$ **to** $n$ **do**
14:     $comp \leftarrow$ RandomChoice($components$)
15:     $dComp \leftarrow$ CreateDistractorComponent($comp$, variation=$i/n$, seed+$i$)
16:     ReplaceComponent($comp$, $dComp$)
17:     RenderImage(q_id_A$\{i\}$)
18:     RestoreComponent($comp$)
19: **end for**
20: SaveMetadata($\{$q_id, $seed$, mainView:$qView$, oppView:$oppView\}$)

---

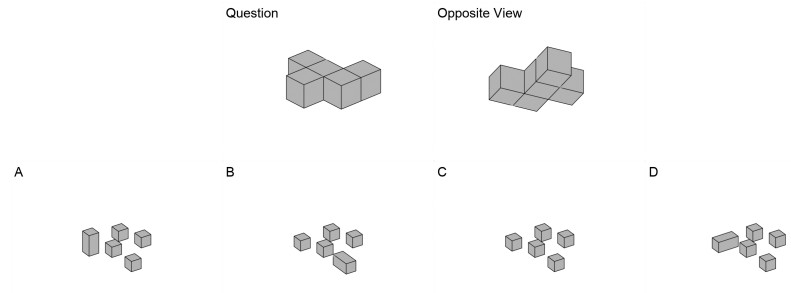

**Figure 7:** Synthetic 3D Combination Data Example.

**3D Combination** The 3D combination task, evaluates the ability to mentally deconstruct a complex 3D object into its constituent parts and then identify which of those parts could be used to build a different target shape.

The task generation proceeds as follows: Shape Generation: A complex 3D shape is created, which serves as the "source" object. This source object is rendered from two opposite isometric viewpoints to give the user a complete understanding of its structure. Component Segmentation: The source object is programmatically broken down into a set of smaller, non-overlapping 3D components. These components are the basic building blocks that could theoretically form the original shape. Question Formulation: The "question" is presented as a new, different "target" 3D shape. Option Generation: The options provided to the user are the individual 3D components that were segmented from the original source object. These components are laid out individually for clear inspection.

---

**Algorithm 4:** GENERATE3DFOLDINGQUESTION

1: **Input:** q_id, difficulty tier list $\{easy, medium, hard\}$, number of distractors $n$
2: $seed \leftarrow \text{Hash}(\text{q\_id})$
3: SetRandomSeed($seed$); ClearScene()
4: $cube, faceMap \leftarrow$ CreateCubeWithTextures(seed=$seed$)
5: $layout \leftarrow RandomChoice(\{cross, T\})$
6: $net \leftarrow UnfoldCube$(cube,layout)
7: SetCamera(Top); RenderImage(q_id_Q)
8: $bestView \leftarrow Best3DView$(cube)
9: $SetCamera$(bestView)
10: RenderImage(q_id_A0)
11: **for** $i \leftarrow 1$ **to** $n$ **do**
12:    $tier \leftarrow$ **SelectTier**$(i, n)$
13:    $dCube \leftarrow$ CreateCubeDistractor($cube$,tier=$tier$, seed+$i$)
14:    SetCamera($bestView$)
15:    RenderImage(q_id_A$\{i\}$)
16: **end for**
17: SaveMetadata($\{$q_id, $seed, layout, faceMap\}$)

---

**3D Folding** The 3D box folding task evaluates a person's spatial reasoning ability, specifically their capacity to visualize how a 2D pattern (a "net") will fold into a 3D cube. The process for generating a question is as follows:

Cube and Texture Generation: A standard 3D cube is created. Each of its six faces is assigned a unique texture or color. This is the "target" cube.

Unfolding: The textured 3D cube is computationally "unfolded" into a 2D net. The net is a flat pattern that shows all six faces of the cube connected in a way that it could be folded back up into the cube. Common net patterns like a "cross" or "T-shape" are used. This 2D net serves as the "question" image.

Option Generation: A set of 3D cubes is then presented as the answer options.

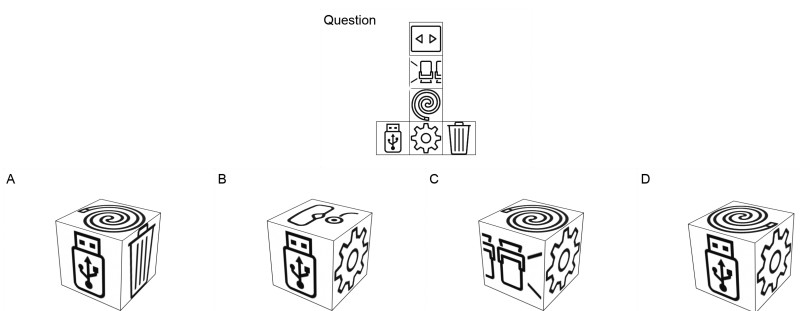

**Figure 8:** Synthetic 3D Folding Data Example.

The Correct Answer: This is a 3D rendering of the original, correctly folded cube, showing how the face textures are oriented in relation to each other.

Distractors: These are 3D cubes that are almost correct but have one or more faces manipulated in a way that makes the folded result incorrect. These manipulations can include:

Face Rotation: One or more faces on the cube are rotated from their correct orientation. Face Swapping: The positions of two or more faces are swapped. Texture/Color Replacement: The texture or color of one face is replaced with that of another.

---

Algorithm 5: GENERATESHAPEFINDINGQUESTION

---

1: **Input:** question id q_id, difficulty $\in \{\text{easy}, \text{medium}, \text{hard}\}$, options $m = 4$
2: $seed \leftarrow \text{Hash}(\text{q\_id})$
3: SetRandomSeed($seed$); ClearScene()
4: $cube \leftarrow CreateCubeWithTextures(seed)$
5: $(V_0, V_1, V_2) \leftarrow ChooseDistinctViews(cube, 3, 120°)$
6: **for** $j \leftarrow 0$ **to** 1 **do**
7:    SetCamera($V_j$), RenderImage(q_id_V{i})
8: **end for**
9: $vis \leftarrow VisibleFaces(cube, V_2)$
10: $f^* \leftarrow SampleFace(vis, \text{strategy}=difficulty)$
11: $mat_{\text{orig}} \leftarrow GetMaterial(cube, f^*)$
12: SetMaterial($cube, f^*, \textit{Blue}$), SetCamera($V_2$), RenderImage(q_id_V2)
13: SetMaterial($cube, f^*, mat_{\text{orig}}$)
14: $opts \leftarrow \{f^*\} \cup Sample(OtherFaces(cube, f^*), m-1)$
15: $opts \leftarrow Shuffle(opts)$
16: **for** $k, f$ **in** Enumerate($opts$) **do**
17:    SetCamera(FaceNormalView(cube,f))
18:    RenderImage(q_id_O{i})
19: **end for**
20: SaveMetadata{id:q_id, $seed$, views:$[V_0, V_1, V_2]$, replaced:$f^*$, correctIdx:$IndexOf(f^*, opts)$}

---

**3D Shape Finding** The 3D Shape Finding task is a visual memory and attention task that tests the ability to track a specific face of a 3D object as the object is rotated in space. Here is how a typical question is generated: Cube Generation: A 3D cube is created with a unique, distinct texture applied to each of its six faces. View Sequence: The participant is shown a sequence of images (typically two) of the cube from different viewpoints. This allows them to see the cube and the arrangement of its face textures from multiple angles. The "Change" Event: A third image of the cube is then presented. In this view, one of the visible faces of the cube has its texture replaced with a solid color (e.g., blue). This is the key event in the task. The Question: The participant is implicitly asked: "Which of the original face textures was replaced by the solid color?" Option Generation: The answer options are a set of images, each showing one of the original, individual face textures from the cube. The Correct Answer: This is the image of the face texture that was replaced by the solid color in the third view. Distractors: These are the other original face textures from the cube.

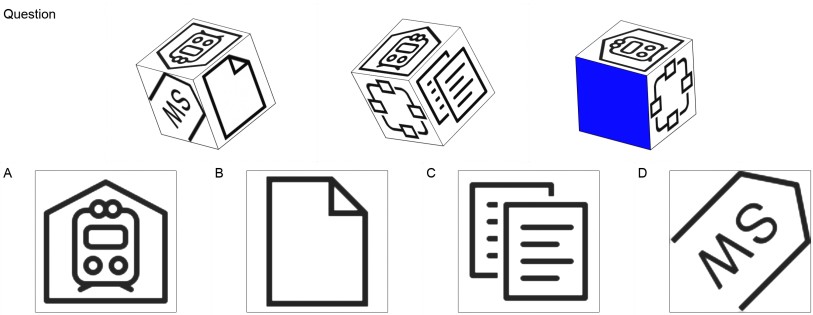

**Figure 9:** Synthetic 3D Shape Folding Data Example.

| Task Name | Distractor Generation Logic | Specific Implementation Details |
|---|---|---|
| 3D Rotation | A new 3D shape is created that is structurally different from the original, yet visually similar, testing the ability to spot subtle structural changes despite viewing-angle differences. | A new shape is generated by altering the "growth history" of the original object (e.g. adding or removing a block in a different location), producing a plausible but incorrect alternative. |
| 3D Projection | An incorrect 2D orthographic projection is generated, testing the ability to accurately project a 3D object onto a 2D plane. | Generating a projection from an incorrect viewpoint (e.g. providing a *side view* when the *top view* was requested). Generating a projection of a slightly modified (distractor) 3D shape. |
| 3D Combination | A valid component from the original shape is structurally modified, testing detailed analysis of part geometry. | A single authentic component is duplicated and then altered—typically by adding or removing a block—yielding a visually similar part that would not fit correctly into the complete assembly. |
| 3D Folding | The 2D net is "folded" into an incorrect 3D cube, testing the ability to track face orientation and adjacency during folding. | **Rotation**: A face's texture is rotated by $90°$, $180°$, or $270°$. **Swapping**: Textures between two faces are swapped. **Flipping**: A texture is flipped horizontally or vertically. |
| 3D Shape Finding | The options presented are the other, non-target faces of the cube, testing visual working memory and attention. | The task is to identify the original texture of a face that was replaced by a solid colour; distractors are the original textures of the other cube faces that were *not* the replacement target. |

**Table 8:** Summary of Distractor Generation Logic

## B  EVALUATION DETAILS

### B.1  HUMAN PERFORMANCE ASSESSMENT DETAILS

To establish a robust human baseline, we recruited 54 participants through a custom online platform. The process yielded 1,684 valid responses, with each of the 559 benchmark items being answered by an average of 3 participants.

The median response time across all human responses was 26.9 seconds, with a mean of 40.3 seconds. The difference suggests that a subset of questions required substantially longer deliberation, skewing the mean. A detailed breakdown of performance by task category, presented in Table 9, reveals a clear inverse relationship between response time and accuracy. The analysis highlights that the two

**Table 9:** Human Performance by Task: Accuracy and Response Time.

| Task | DISE Category | Accuracy (%) | Mean Time (s) | Median Time (s) |
|------|---------------|--------------|---------------|-----------------|
| 3D Combination | E–D | 56.4 | 59.2 | 34.8 |
| 2D Shape Finding | I–S | 61.5 | 58.5 | 46.4 |
| 2D Combination | E–D | 75.2 | 36.8 | 32.0 |
| 2D Folding | I–D | 76.5 | 25.6 | 17.0 |
| Fold and Punch | I–D | 76.8 | 55.4 | 44.4 |
| 2D Rotation | I–D | 78.1 | 40.4 | 31.5 |
| 3D Projection | E–S | 81.1 | 28.0 | 20.8 |
| 3D Shape Finding | I–S | 81.8 | 31.4 | 23.3 |
| 3D Rotation | I–D | 82.0 | 29.8 | 21.8 |
| 3D Folding | I–D | 86.6 | 44.4 | 33.6 |

tasks with the lowest human accuracy, 3D Combination (56.4%) and 2D Shape Finding (61.5%), are also the tasks that commanded the longest mean response times (59.2s and 58.5s, respectively). This empirically confirms that these tasks impose the highest cognitive load. The mental simulation required to assemble complex parts in 3D Combination (Extrinsic-Dynamic) and the demanding visual search needed to disentangle embedded figures in 2D Shape Finding (Intrinsic-Static) are inherently time-consuming and error-prone for humans, providing a quantitative justification for their difficulty. Conversely, tasks with high accuracy, such as 3D Folding and 3D Rotation, generally required less time, indicating a lower cognitive barrier.

In order to obtain an unbiased estimate of human baseline performance over the full item pool, we employ a matrix-sampling design in which each participant completes only a single booklet of $K$ items out of the total pool of $I$ items. Adjacent booklets share a small set of $a$ anchor items ($\approx$10 We recruited 54 participants for the study. Before participation, all individuals provided their informed consent and all procedures were conducted in accordance with relevant ethical guidelines. The data collection process yielded a total of 1679 valid responses across all items. Each item was answered by an average of 3 participants. The main paper reports human performance with Classical Test Theory (CTT) results, while Item Response Theory (IRT) is used for cross-validation.

**Analysis Methodology**  The collected response data were analyzed using two psychometric frameworks:

CLASSICAL TEST THEORY (CTT)  For each item booklet and for the anchor-linked "overall" pool, the proportion-correct statistic was computed as

$$\hat{p} = \frac{x}{N} \tag{1}$$

where x is the number of correct responses and N is the total number of responses to that booklet or pool. Sampling variability was quantified with the Wald standard error

$$\text{SE}_{\text{CTT}} = \sqrt{\frac{\hat{p}(1-\hat{p})}{N}}, \tag{2}$$

yielding a two-sided 95% confidence interval (CI)

$$\hat{p} \pm 1.96 \, \mathrm{SE_{CTT}}. \tag{3}$$

ITEM RESPONSE THEORY (IRT)  To cross-validate the CTT findings and place all items on a common latent-ability scale, we fitted a two-parameter logistic (2PL) model to the entire response matrix,

$$P_{ij} = \sigma\big[a_i\,(\theta_j - b_i)\big] = \frac{1}{1 + \exp\big[-a_i(\theta_j - b_i)\big]}, \tag{4}$$

where $P_{ij}$ is the probability that participant j (ability $\theta_j$) answers item i (discrimination $a_i$, difficulty $b_i$) correctly.

For a designated item subset (e.g., a DISE category) containing $I$ items, the model yields an item-level expected probability of success $\bar{P}_i$. The category-level expected accuracy is then

$$\hat{p}_{\mathrm{IRT}} = \frac{1}{I} \sum i = 1^I \bar{P}_i. \tag{5}$$

Between-item variability was captured via the sample variance

$$s^2 = \frac{1}{I-1} \sum_{i=1}^{I} \big(\bar{P}i - \hat{p}\mathrm{IRT}\big)^2, \tag{6}$$

leading to the standard error

$$\mathrm{SE_{IRT}} = \frac{s}{\sqrt{I}}, \tag{7}$$

and the 95% CI

$$\hat{p}_{\mathrm{IRT}} \pm 1.96 \, \mathrm{SE_{IRT}}. \tag{8}$$

**Results**  To provide a comprehensive view, we compare the results from both CTT and IRT analyses. Figure 10, Table 11 juxtaposes the observed accuracy from CTT with the model-based predictions and item parameters from IRT for each DISE category. This comparison highlights the synergy between the two methodologies. The CTT accuracy provides a direct, empirical measure of performance, while the IRT parameters offer an explanation for these results.

Table 10: Parameters of the Human Assessment

| Parameters | Num. |
|---|---|
| Number of Participants | 54 |
| Total Number of Items $I$ | 559 |
| Total Responses $N$ | 1679 |
| Number of Booklets | 19 |

Table 11: Human Accuracy by DISE Category

| DISE Category | CTT Accuracy (95% CI) | IRT Accuracy (95% CI) |
|---|---|---|
| Extrinsic–Dynamic | 61.05% ± 4.46% | 57.22% ± 8.67% |
| Extrinsic–Static | 81.12% ± 4.38% | 82.09% ± 7.02% |
| Intrinsic–Dynamic | 80.25% ± 2.05% | 81.09% ± 3.42% |
| Intrinsic–Static | 76.80% ± 3.90% | 77.29% ± 7.49% |
| Overall | 76.84% ± 2.02% | 76.92% ± 3.79% |

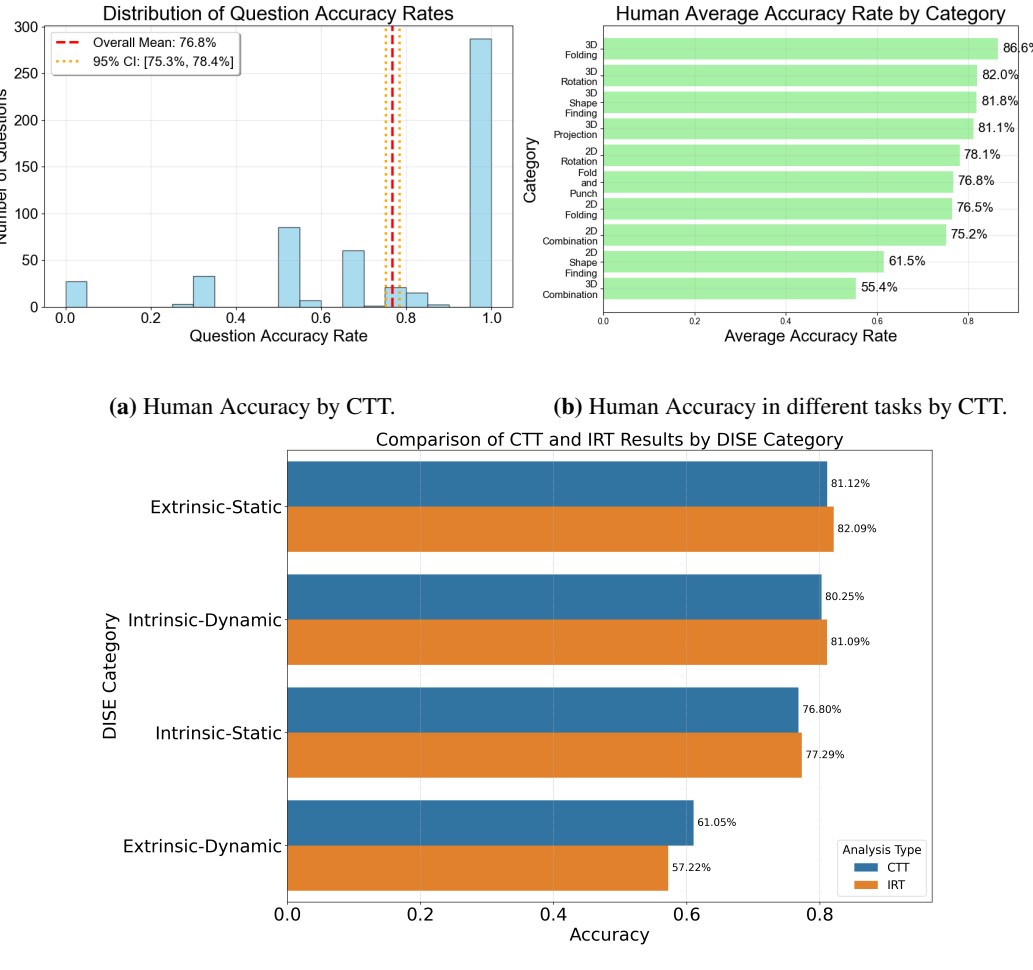

(a) Human Accuracy by CTT.  (b) Human Accuracy in different tasks by CTT.

(c) Comparison of CTT and IRT results.

**Figure 10:** Human Performance Results by CTT and comparison of CTT and IRT results.

## B.2 Evaluation Implementation Details

All evaluations were implemented on 3 NVIDIA A100-40G with VLMEvalKit v0.2. Following the idea of Duan et al. (2025), all the models used very low temperatures or temperatures equal to 0 and set $do\_sample = False$ to ensure reproducibility and certainty of the results. The API checkpoints for proprietary models are listed in Table 12.

**Table 12:** Proprietary APIs evaluated in this paper

| Proprietary Model & provider | API endpoint |
| --- | --- |
| Claude 3.7 Sonnet (Anthropic) | `claude-3-7-sonnet-20250610` |
| Doubao 1.5 VL (volcengine) | `doubao-1-5-vision-pro-32k-250115` |
| Doubao 1.5 VL-thinking (volcengine) | `doubao-1-5-thinking-vision-pro-250428` |
| Gemini 2.0 Flash (Google) | `gemini-2.0-flash` |
| Gemini 2.5 Flash (Google) | `gemini-2.5-flash-preview-05-20` |
| Gemini 2.5 Flash w/o thinking (Google) | `gemini-2.5-flash-preview-05-20` |
| GPT-4.1 nano (OpenAI) | `gpt-4.1-nano-2025-04-14` |
| GPT-4o (OpenAI) | `gpt-4o-2024-08-06` |
| GPT-4o-mini (OpenAI) | `gpt-4o-mini-2025-06-10` |
| GPT-5 (OpenAI) | `gpt-5` |
| o4-mini (OpenAI) | `o4-mini-2025-04-16` |

The prompt templates used in the evaluation for different models are shown below:

**Listing 1:** Prompt Templates used for Proprietary Models in VLMEvalKit

```
PROMPT_TEMPLATES = {
    "SYSTEM": "You are a helpful assistant.",

    "USER": """<image>
    Question: The two images above show a 3D structure from
    different angles. Which one of the options below could be
    constructed to appear the same as both given views when observed
    from the corresponding perspectives without rotation and overlaps?
    Select the most likely one.
        Options:
        A. A
        B. B
        C. C
        D. D
        Answer with the option's letter from the given choices
    directly."""
}
```

**Listing 2:** Prompt Templates used for Llama Serie Models in VLMEvalKit

```
PROMPT_TEMPLATES = {
    "SYSTEM": "",

    "USER": """<|begin_of_text|><|start_header_id|>user<|end_header_id|>

<im_start><image><im_end>
        Question: The two images above show a 3D structure from
    different angles. Which one of the options below could be
    constructed to appear the same as both given views when observed
    from the corresponding perspectives without rotation and overlaps?
    Select the most likely one.
        Options:
        A. A
```

```
        B. B
        C. C
        D. D
Answer with the option's letter from the given choices
    directly.<|eot_id|>"""
}
```

**Listing 3:** Prompt Templates used for QwenVL, InternVL, Ovis2 Serie Models in VLMEvalKit

```
PROMPT_TEMPLATES = {
    "USER": """<image>
        Question: The two images above show a 3D structure from
    different angles. Which one of the options below could be
    constructed to appear the same as both given views when observed
    from the corresponding perspectives without rotation and overlaps?
    Select the most likely one.
        Options:
        A. A
        B. B
        C. C
        D. D
        Please select the correct answer from the options above."""
}
```

**Listing 4:** Prompt Templates used for VLM-R1 and LMM-R1 in VLMEvalKit

```
PROMPT_TEMPLATES = {
    "USER": """<image>
        Question: The two images above show a 3D structure from
    different angles. Which one of the options below could be
    constructed to appear the same as both given views when observed
    from the corresponding perspectives without rotation and overlaps?
    Select the most likely one.
        Options:
        A. A
        B. B
        C. C
        D. D
        Please select the correct answer from the options above. Output
    the thinking process in <think> </think> and final answer in
    <answer> </answer> tags."""
}
```

**Listing 5:** Prompt Templates used for VLAA_Thinker Serie Models in VLMEvalKit

```
PROMPT_TEMPLATES = {
    "SYSTEM": "You are VL-Thinking, a helpful assistant with excellent
    reasoning ability. You should first think about the reasoning
    process and then provide the answer. Use <think>...</think> and
    <answer>...</answer> tags."

    "USER": """<image>
        Question: The two images above show a 3D structure from
    different angles. Which one of the options below could be
    constructed to appear the same as both given views when observed
    from the corresponding perspectives without rotation and overlaps?
    Select the most likely one.
        Options:
        A. A
        B. B
        C. C
        D. D
        Please select the correct answer from the options above."""
}
```

## B.3 MORE EVALUATION RESULTS

**Table 13:** Different-task accuracies on Spatial-DISE Bench. Abbreviations—2D Comb.: 2D Combination; 2D Fold.: 2D Folding; 2D Rot.: 2D Rotation; 2D S.F.: 2D Shape Finding; 3D Comb.: 3D Combination; 3D Fold.: 3D Folding; 3D Proj.: 3D Projection; 3D Rot.: 3D Rotation; 3D S.F.: 3D Shape Finding; F&P: Fold and Punch. **Bold** indicates the highest accuracy; Underline indicates the second highest.

| Model | Acc. | 2D Comb. | 2D Fold. | 2D Rot. | 2D S.F. | 3D Comb. | 3D Fold. | 3D Proj. | 3D Rot. | 3D S.F. | F&P |
|---|---|---|---|---|---|---|---|---|---|---|---|
| *Proprietary* | | | | | | | | | | | |
| Claude 3.7 Sonnet | 30.6% | 29.2% | 25.6% | 30.4% | **38.1%** | 20.0% | 50.7% | 31.4% | 31.4% | 28.8% | 24.4% |
| Doubao1.5 VL | 33.8% | **41.7%** | 25.6% | **43.5%** | 33.3% | 26.7% | 44.9% | **37.1%** | 37.1% | **34.8%** | 25.6% |
| Gemini 2.0 Flash | **34.2%** | 20.8% | **41.0%** | 30.4% | 23.8% | 26.7% | 56.5% | 31.4% | **51.4%** | 15.2% | 24.4% |
| GPT4.1 nano | 29.3% | 29.2% | **35.9%** | 30.4% | 14.3% | **30.0%** | 36.2% | 35.7% | 30.0% | 18.2% | 23.1% |
| GPT4o | 28.1% | 29.2% | 26.9% | 17.4% | 33.3% | 25.0% | 30.4% | 22.9% | 32.9% | 25.8% | **33.3%** |
| GPT4o-mini | 25.6% | 20.8% | 28.2% | 30.4% | 28.6% | 15.0% | 37.7% | 21.4% | 22.9% | 28.8% | 23.1% |
| Gemini 2.5 Flash | 31.5% | 12.5% | 33.3% | 17.4% | 33.3% | 18.3% | 69.6% | 27.1% | 40.0% | 16.7% | 24.4% |
| Gemini 2.5 Flash w/o thinking | 32.0% | 12.5% | 33.3% | 17.4% | 33.3% | 16.7% | 69.6% | 28.6% | 40.0% | 19.7% | 25.6% |
| GPT-5 | 30.1% | 20.8% | 30.8% | 43.5% | 14.3% | 25.0% | 31.9% | 25.7% | 45.7% | 30.3% | 24.4% |
| o4-mini | 33.3% | 33.3% | 30.8% | 52.2% | 23.8% | 10.0% | 47.8% | 25.7% | 38.6% | 48.5% | 26.9% |
| *Proprietary Average* | *30.9%* | *25.0%* | *31.1%* | *31.3%* | *27.6%* | *21.3%* | *47.5%* | *28.7%* | *37.0%* | *26.7%* | *25.5%* |
| *Open-source* | | | | | | | | | | | |
| Llama-3V-11B | 24.5% | 29.2% | 24.4% | 21.7% | 19.0% | 30.0% | 31.9% | 14.3% | 24.3% | 25.8% | 23.1% |
| Cambrian-13b | 26.7% | 20.8% | 30.8% | 30.4% | 23.8% | 26.7% | 21.7% | 32.9% | 25.7% | 27.3% | 23.1% |
| Cambrian-8b | 22.9% | 25.0% | 26.9% | 30.4% | **33.3%** | 16.7% | 33.3% | 15.7% | 15.7% | 27.3% | 17.9% |
| InternVL3-38B | **32.4%** | 29.2% | 28.2% | **47.8%** | 23.8% | 26.7% | 42.0% | 30.0% | 40.0% | 27.3% | **30.8%** |
| InternVL3-14B | 31.1% | 25.0% | 24.4% | 21.7% | 14.3% | 20.0% | **53.6%** | 31.4% | **42.9%** | 19.7% | **34.6%** |
| InternVL3-8B | 26.3% | **33.3%** | **35.9%** | 30.4% | 14.3% | 20.0% | 29.0% | 28.6% | 32.9% | 9.1% | 25.6% |
| Kimi-VL-A3B | 24.3% | 12.5% | 29.5% | 26.1% | **33.3%** | 20.0% | 26.1% | 27.1% | 35.7% | 10.6% | 20.5% |
| Ovis2-16B | 26.3% | 20.8% | 16.7% | 13.0% | 19.0% | 20.0% | 52.2% | 27.1% | **42.9%** | 10.6% | 23.1% |
| Ovis2-8B | 23.8% | 25.0% | 28.2% | 17.4% | 28.6% | 11.7% | 36.2% | 21.4% | 34.3% | 7.6% | 24.4% |
| Qwen2.5-VL-32B | 27.2% | 20.8% | 19.2% | 21.7% | 21.7% | 21.7% | 34.8% | 31.4% | 35.7% | **28.8%** | 24.4% |
| Qwen2.5-VL-7B | 26.1% | **33.3%** | 26.9% | 39.1% | **33.3%** | **31.7%** | 30.4% | 24.3% | 32.9% | 10.6% | 17.9% |
| Qwen2.5-VL-3B | 22.9% | 29.2% | 28.2% | 17.4% | 14.3% | 23.3% | 36.2% | 17.1% | 22.9% | 12.1% | 21.8% |
| *Open-source Average* | *26.2%* | *25.3%* | *26.6%* | *26.4%* | *23.4%* | *22.4%* | *35.6%* | *25.1%* | *32.2%* | *18.1%* | *23.9%* |
| *Reasoning & Spatial-Specified Models* | | | | | | | | | | | |
| LLaVA-CoT | 24.0% | 29.2% | **34.6%** | 13.0% | 9.5% | 30.0% | 17.4% | 22.9% | 22.9% | 19.7% | 25.6% |
| LMM-R1 | 26.1% | 29.2% | 28.2% | 21.7% | 38.1% | 30.0% | 36.2% | 20.0% | 24.3% | 22.7% | 19.2% |
| VLM-R1 | 30.8% | 25.0% | 26.9% | 39.1% | 38.1% | 36.7% | 47.8% | 18.6% | 30.0% | 24.2% | 29.5% |
| Kimi-VL-A3B-Thinking | 24.7% | 16.7% | 26.9% | 26.1% | **42.9%** | 31.7% | 26.1% | 28.6% | 22.9% | 15.2% | 19.2% |
| Doubao1.5-VL-thinking | 42.0% | **62.5%** | 28.2% | **43.5%** | 23.8% | **61.7%** | 56.5% | 31.4% | **50.0%** | **39.4%** | 30.8% |
| VLAA-Thinker-3B | 25.9% | 37.5% | 20.5% | 26.1% | 28.6% | 25.0% | 36.2% | 30.0% | 27.1% | 9.1% | 28.2% |
| VLAA-Thinker-7B | 27.9% | 25.0% | 25.6% | 26.1% | 38.1% | 28.3% | 31.9% | 27.1% | 35.7% | 22.7% | 23.1% |
| SpaceThinker | 32.6% | 29.2% | 20.5% | **43.5%** | 33.3% | 43.3% | 49.3% | 22.9% | 35.7% | 22.7% | 33.3% |
| SpaceOm | 25.9% | 25.0% | 14.1% | **43.5%** | 33.3% | 36.7% | 49.3% | 24.3% | 32.9% | 24.2% | **37.2%** |
| SpaceR | 27.0% | 37.5% | 32.1% | 34.8% | 28.6% | 26.7% | 29.0% | 17.1% | 37.1% | 21.2% | 19.2% |
| *Reasoning & Spatial Average* | *27.6%* | *27.8%* | *25.9%* | *28.6%* | *27.0%* | *28.2%* | *37.1%* | *25.1%* | *31.8%* | *19.8%* | *25.4%* |
| *Overall Average* | *28.4%* | *27.2%* | *27.8%* | *29.6%* | *27.2%* | *26.0%* | *40.1%* | *26.0%* | *33.6%* | *22.0%* | *25.2%* |
| SpaceOm-sft | 33.8% | 25.0% | 25.6% | 26.1% | 23.8% | 45.0% | 46.4% | 31.4% | 50.0% | 30.3% | 20.5% |
| Qwen2.5-VL-7B-sft | 47.0% | 33.3% | 34.6% | 34.8% | 9.5% | 78.3% | 69.6% | 41.4% | 65.7% | 69.7% | 21.8% |
| *Human* | *76.8%* | *75.2%* | *76.5%* | *78.1%* | *61.5%* | *55.4%* | *86.6%* | *81.1%* | *82.0%* | *81.8%* | *76.8%* |

## B.4 SUPERVISED FINE-TUNING HYPERPARAMETERS

The Supervised Fine-Tuning (SFT) experiments were conducted using the Swift framework. We employed the Low-Rank Adaptation (LoRA) technique to efficiently fine-tune both the Qwen2.5-VL-7B and SpaceOm models on the Spatial-DISE-12K training set. All linear layers of the models were targeted for LoRA adaptation. The key hyperparameters used for the fine-tuning process are detailed in Table 13.

**Table 14:** Hyperparameters for SFT Training

| Hyperparameter | Value |
|---|---|
| Framework | Swift |
| Fine-Tuning Method | LoRA |
| Target Modules | all-linear |
| LoRA Rank (lora_rank) | 8 |
| LoRA Alpha (lora_alpha) | 32 |
| Batch Size | 48 |
| Precision (torch_dtype) | bfloat16 |
| Learning Rate | 1.5e-4 |
| Warmup Ratio | 0.05 |
| Number of Epochs | 2 |
| Max Sequence Length | 4096 |
| Deepspeed | zero3 |

# C   ERROR ANALYSIS DETAILS

This section provides a detailed quantitative and qualitative breakdown of the error analysis conducted to understand the failure modes of VLMs on Spatial-DISE Bench.

## C.1   DEFINITION OF HIGH-LEVEL ERROR

We established a high-level error taxonomy to systematically diagnose failures by deconstructing the model mistakes into three errors:

- **Perceptual Error**, which the model fails to accurately interpret basic visual information, such as the shape, count, or spatial relationship of objects.
- **Comprehension Error**, which the model misinterprets the natural language prompt or the objective of the task, indicating a failure to understand the question.
- **Reasoning Error**, which the model correctly perceives the visual scene and understands the prompt but fails in the logical deduction required to reach the correct answer. This includes errors in mental rotation, folding, or spatial manipulation.

## C.2   VLM-AS-JUDGE IN ERROR ANALYSIS

Inspired by Yang et al. (2025), we adopted an automated error analysis pipeline. As shown in Figure 11, we use Doubao-1.6-thinking as a judge, combined with human inspection. Table 15 lists the distribution of the wrong responses sampled in error analysis.

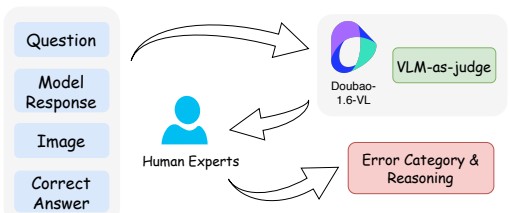

**Table 15:** Error Distribution by DISE category

| DISE Category | Count |
|---|---|
| Intrinsic-Static (I-S) | 34 |
| Intrinsic-Dynamic (I-D) | 107 |
| Extrinsic-Static (E-S) | 34 |
| Extrinsic-Dynamic (E-D) | 25 |
| **Total** | **200** |

**Figure 11:** Error Analysis Pipeline

The prompt template used for error analysis:

**Listing 6:** Prompt Templates used for Error Analysis

```
    ERROR_ANALYSIS_PROMPTS = {
    "detailed_analysis": """
Please provide a detailed analysis of the visual-language model's
    incorrect answer:

Question Category: {category}
Question: {question}

Options:
{options}

Correct Answer: {correct_answer}
Model's Predicted Answer: {predicted_answer}
Model's Full Response: {model_prediction}

Please analyze in depth from the following perspectives:
1. Error Type Classification:
   - Perception Error: The model failed to correctly identify visual
    elements in the image.
   - Comprehension Error: The model recognized visual elements but
    misunderstood their meaning.
   - Reasoning Error: The model understood the content but made a
    mistake in reasoning.
```

```
2. Specific Cause of the Error
3. Severity Assessment (Low / Medium / High)
4. Possible Directions for Improvement
5. Suggestions to Prevent Similar Errors

Please return the analysis in JSON format:
{{
    "Error Type": "Specific type of error",
    "Error Subtype": "More detailed category of the error",
    "Cause of Error": "Detailed explanation of the cause",
    "Severity": "Low/Medium/High",
    "Summary": "Brief summary of the error"
}}
""",

    "category_analysis": """
Please analyze the error patterns of the visual-language model in the
    following {category} category questions:

{error_examples}

Analyze from the following perspectives:
1. Most common error types in this category
2. Common features and patterns of errors
3. Category-specific challenges

Please provide a structured response.
""",

    "comparison_analysis": """
Please compare the error performance of the following models on the same
    question:

{model_comparisons}

Analyze:
1. Differences in error types across models
2. Strengths and weaknesses of each model
3. Comparison of error severity

Please provide a detailed comparative analysis.
"""
}
```

Our analysis reveals a clear and consistent pattern: Reasoning Error is the predominant failure category, accounting for an overwhelming 72.5% (145 out of 200) of all analyzed mistakes. Perceptual errors constituted 17.5% of the total, while comprehension errors were the least common at 10%. This distribution strongly suggests that the primary bottleneck for current VLMs is not in visual perception but in complex spatial-logical inference. While this initial classification identifies where the models fail, a more granular analysis is required to understand why they fail.

## C.3  A DEEP DIVE INTO REASONING FAILURES

To move from symptom to cause, we performed a deeper analysis of the 145 reasoning errors, re-categorizing them based on the underlying cognitive abilities that are deficient. This approach, inspired by cognitive science, reveals that the models' failures stem from a lack of fundamental cognitive mechanisms for spatial intelligence. We identified three primary root causes.

**Failure in Rule Application (44.8%)**  This was the most critical category of failure. Models demonstrate an ignorance of the fundamental axioms, constraints, and invariances of the geometric world. The errors are not in complex derivations but in the application of basic, non-negotiable

rules. The root cause appears to be a failure to link visual percepts to an abstract library of geometric principles; the models see pixels, not entities governed by rules.

A frequent failure was confusing adjacent and opposite faces in 3D cube problems. For instance, a model might correctly identify the symbols on a cube's faces but fail to apply the simple rule that adjacent faces cannot be opposite one another.

**Failure in Mental Simulation (40.0%)** The second most significant failure was the inability to construct a dynamic, operable internal representation to simulate a continuous spatial transformation. Models lack a reliable "spatial working memory" to track an object's state through a sequence of operations. They cannot robustly answer the question, "what happens next?"

This was most evident in "Fold and Punch" tasks. Models consistently failed to track the number of layers created by folds and, consequently, could not predict the symmetric replication of holes upon unfolding. For example, after simulating a two-fold process (creating four layers), a model might incorrectly predict only two holes in the unfolded paper, demonstrating a breakdown in state tracking.

**Failure in Holistic-Local Processing (15.2%)** Finally, models exhibited an imbalance in processing visual information, struggling to shift between holistic understanding and local detail analysis. Their attention mechanisms appear unable to dynamically allocate cognitive resources to the most salient features required by the task.

Models were often misled by superficial similarity. In rotation tasks, a model might identify an option as correct simply because it "looks similar" to the target, while ignoring a fatal flaw in the local arrangement of its components, such as an incorrect orientation of a key part.

In summary, the failures of current VLMs in spatial reasoning are systemic and deeply rooted in cognitive deficiencies. They lack an internal "world model" that is constrained by geometric rules, can be manipulated through dynamic simulation in a spatial working memory, and is guided by a flexible attentional mechanism. This points to a clear direction for future research: efforts must transcend simple pattern matching and focus on imbuing models with the foundational capabilities for genuine spatial cognition.

**Table 16:** Error Analysis Across Different Models

| Err.\ Models | Qwen2.5-VL | GeminiFlash | Doubao-1.5 | SpaceThinker |
|---|---|---|---|---|
| Reasoning Err. | 31 | 31 | 37 | 46 |
| Perceptual Err. | 12 | 12 | 8 | 3 |
| Comprehension Err. | 7 | 7 | 5 | 1 |

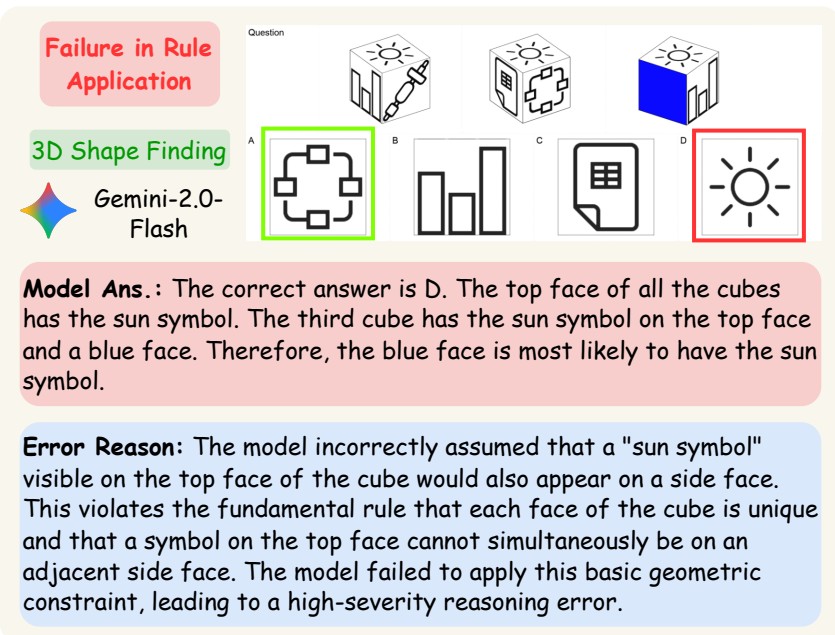

**Figure 12:** Error example of Failure in Rule Application

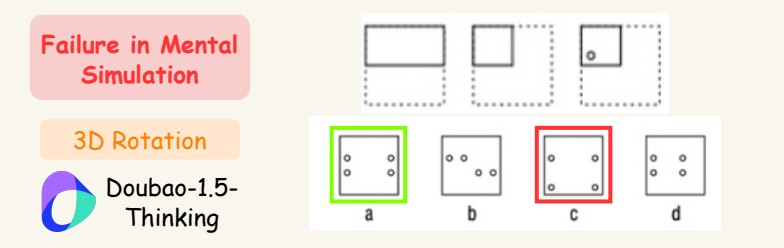

**Failure in Mental Simulation**

**3D Rotation**

Doubao-1.5-Thinking

**Model Ans.:** To solve this problem, we analyze the folding and punching process:
1. Folding Steps: The square is folded twice. First, a horizontal fold (reducing height), then a vertical fold (reducing width). This creates 4 layers of the square.
2. Punching a Hole: When a hole is punched through 4 layers, the unfolded square will have 4 holes (one in each layer).
3. Position of Holes: The hole in the folded square is symmetric across both folds. When unfolded, the holes must be evenly distributed in a symmetric pattern.
Among the options:
- a and d have only 2 holes (insufficient, as 4 layers mean 4 holes).
- b has 4 holes but their placement is not symmetric with the folding logic.
- c has 4 holes arranged symmetrically, matching the result of two folds and a single punch.
Thus, the correct answer is c.

**Error Reason:** The model correctly identified that the process would result in four holes. However, it failed to accurately simulate the unfolding process. It incorrectly calculated the reflective positions of the holes, placing them on the middle and bottom edges instead of in the middle. This demonstrates a breakdown in its ability to track the state and symmetric transformations of an object through a multi-step process.

**Figure 13:** Error example of Failure in Mental Simulation

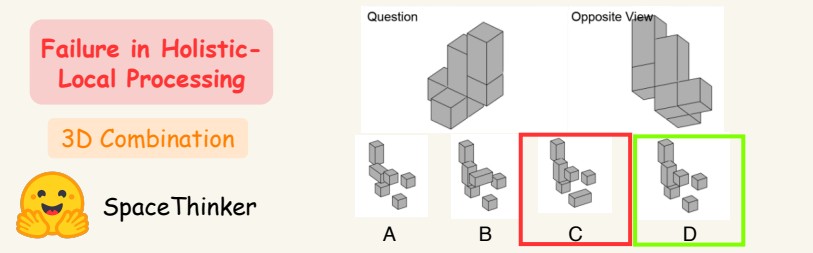

**Failure in Holistic-Local Processing**

**3D Combination**

😄 SpaceThinker

**Model Ans.:**
<think>
To determine the correct answer, we need to analyze the two perspectives and find a combination that matches both views without rotation or overlaps.
1. **Question View**: The structure in the question view shows a central column with smaller blocks around it.
2. **Opposite View**: The structure in the opposite view shows a similar central column but with additional blocks on the sides.
Looking at the options:
- **Option A**: This arrangement does not match the central column structure in both views.
- **Option B**: This arrangement also does not match the central column structure in both views.
- **Option C**: This arrangement matches the central column structure in both views.
- **Option D**: This arrangement does not match the central column structure in both views.
Given that Option C matches the central column structure in both views without any overlaps or rotations, it is the most likely correct answer.
</think>
<answer>C</answer>

**Error Reason:** Although the model correctly identified the "central pillar structure" in both views, it failed to analyze the detailed spatial configuration of the surrounding blocks, such as their exact positions and quantities. As a result, it could not accurately determine whether an option satisfied the requirements of "no rotation" and "no overlap." Specifically, Option C's surrounding block layout contradicts the details in one of the views (e.g., the block distribution is inconsistent with the perspective), whereas the correct Option D matches the spatial details in both views. However, due to imprecise spatial reasoning, the model incorrectly judged C as valid.

**Figure 14:** Error example of Failure in Holistic-Local Processing

