# OpenReview forum: "Spatial-DISE: A Unified Benchmark for Evaluating Spatial Reasoning in Vision-Language Models"
_ICLR.cc/2026/Conference — ICLR 2026 Poster_

### Official Review · Reviewer_YgFh · 2025-10-25

**Soundness:** 2
**Presentation:** 3
**Contribution:** 2
**Rating:** 4
**Confidence:** 3

**Summary:**

The paper introduces Spatial-DISE, a new benchmark and dataset designed to evaluate the spatial reasoning capabilities of VLMs, along with the data generation pipeline. The authors argue that existing benchmarks are inadequate, often focusing on static scenarios and lacking a systematic, cognitively-grounded framework. To address this, they propose the DISE taxonomy, which categorizes spatial reasoning tasks into four quadrants based on two dimensions: Intrinsic vs. Extrinsic and Static vs. Dynamic. Experiment results show that current models struggle in mental simulation.

**Strengths:**

1. Comprehensive empirical evaluation. The paper presents an extensive and rigorous evaluation across 28 distinct Vision-Language Models. The selection of models is commendable, covering a wide spectrum of architectures including proprietary APIs, open-source foundation models, and specialized models fine-tuned for reasoning. The experiments with fine-tuning in Section 4.3 are interesting, where performance gains on in-domain tasks come at the cost of generalization to other spatial benchmarks.
2. Cognitively grounded and unified taxonomy, which provides a comprehensive analysis framework in analysing spatial reasoning performance of VLMs.
3. The paper is well-written, with rich error analysis.

**Weaknesses:**

1. Fail to disentangle perceptual confounders from reasoning deficits. While the paper attributes model failures primarily to reasoning, as far as I'm concerned, the analysis may not fully account for more fundamental perceptual challenges that act as confounding variables. The evaluation format often presents a complex multi-panel image containing both the question pattern and several graphical options. A model's failure could stem from an inability to correctly understanding this layout which is a visual parsing issue, not a spatial reasoning deficit (which I believe is not really the perceptual error and comprehension error in L447. Please correct me if I'm wrong). This potential confounder weakens several key conclusions:
* The claim about the relationship between static and dynamic reasoning could be affected.
* The "catastrophic forgetting" observed in Section 4.3 could be reinterpreted. The performance drop on other benchmarks might not be a loss of reasoning ability, but rather a failure to generalize to different visual formats and layouts on the perception side.
2. Concerns in task categorization. While the DISE taxonomy is well-founded, the mapping of specific tasks to its quadrants can be ambiguous. A task may not be "cognitively pure" and could be solvable via multiple reasoning strategies that cross quadrant boundaries. For example, 3D Shape Finding is classified as Intrinsic-Static, implying a logical deduction about a fixed object's properties. However, a viable and intuitive strategy to solve this task involves mentally rotating the cube (an Intrinsic-Dynamic process) to build a complete 3D mental model before identifying the missing face. This ambiguity raises concern regarding the diagnostic clarity of the benchmark. If a model fails this task, it is unclear whether the failure lies in static deduction or dynamic mental simulation, which in turn limits the precision of the paper's conclusions about specific cognitive weaknesses.
3. Lack of deeper analysis on model architecture. The evaluation is comprehensive in its breadth of models but could be deeper in its analysis of architectural influences. The paper groups models by family or purpose (e.g., "reasoning models") but does not delve into how specific architectural choices (e.g., type of vision encoder, cross-attention mechanism, size of the vision model vs. language model) might correlate with performance on the different DISE quadrants. (minor comments that are not factored into my score fyi)

**Questions:**

See as above in weaknesses. I'm happy to adjust the scores if the author can address the three concerns in the weaknesses.

Typo: missing . in L433

---

> ### Author Response · Authors · 2025-11-20
>
> We truly appreciate your insightful observation. Below we address each concern and outline the clarifications and revisions we will make.
>
> ### **1. Disentangling perceptual confounders from reasoning deficits**
>
> Thank you for raising this concern. We agree that some errors could, in principle, arise from low-level perceptual challenges rather than genuine spatial reasoning deficits, which is why our analysis explicitly distinguishes **Perceptual**, **Comprehension**, and **Reasoning** errors instead of attributing all failures to “reasoning.”
>
> As defined in Appendix C.1:
>
> - **Perceptual Error** – the model fails to interpret basic visual information (shape, count, or spatial relationships of objects).
> - **Comprehension Error** – the model misinterprets the natural language prompt or task objective.
> - **Reasoning Error** – the model correctly perceives the visual scene and understands the prompt but fails in the logical deduction, including mental rotation, folding, or spatial manipulation.
>
> In our manual analysis of 200 failed responses, Perceptual + Comprehension Errors account for only a **minority** of cases, while the **majority** are annotated as Reasoning Errors. Qualitatively, we observed from responses of reasoning models that models can usually distinguish different panels (e.g. Figure 14 in Appendix C.3) and focus on the correct ones but fail to maintain **3D adjacencies, face correspondences, or global consistency**. In short, the layout is typically understood; what breaks is the spatial reasoning on top of it.
>
> To check whether our conclusions might still be an artifact of the **visual format**, we conducted an ablation that changes only the presentation form while keeping content and semantics fixed. In the standard setting, the question and options are given as one merged image; in the ablation, the question and each option are provided as separate images. For Qwen2.5-VL-7B, accuracy is almost unchanged:
>
> > Merged image: 26.1% vs. Separate images: 24.9%
> >
>
> If the multi-panel layout were the main bottleneck, we would expect a clear improvement in the separate-images condition. The near-identical performance instead suggests that the dominant difficulty lies in **spatial reasoning**, not format parsing.
>
> We also revisited the “catastrophic forgetting” concern by re-running DISE SFT on SpaceOm and evaluating it on multiple **external spatial benchmarks** with different formats:
>
> |  | Spatial-DISE | CVBench | SAT | SPACE | OmniSpatial | VSIBench_MCQ |
> | --- | --- | --- | --- | --- | --- | --- |
> | SpaceOm | 25.9% | 68.8% | 46.67% | 27.22% | 27.91% | 31.05% |
> | +DISE SFT | 41.3% | 70.33% | 49.33% | 32.6% | 34.28% | 33.7% |
> | delta | +15.4% | +1.53% | +2.66% | +5.38% | +6.37% | +2.65% |
>
> Rather than catastrophic forgetting, we observe **consistent gains** across all five external benchmarks. This is hard to reconcile with a purely format-driven explanation and instead suggests that DISE SFT strengthens **spatial reasoning skills** that transfer across different visual formats.
>
> Overall, the error breakdown, qualitative examples, and new experiments indicate that while some perceptual and comprehension errors do occur, our main conclusions are **not** driven by multi-panel format confounders. The dominant bottleneck lies in complex spatial-logical inference, particularly in 3D structure and adjacency reasoning.

---

> > ### Author Response · Authors · 2025-11-20
> >
> > ### **2. Concerns in task categorization under the DISE taxonomy**
> >
> > Thank you for pointing this insightful concern. We agree that a single spatial reasoning task can be solved using different strategies. Humans may switch between **logical deduction**, **mental rotation**, and **holistic mental modeling** even on the same item.
> >
> > DISE taxonomy, however, is intentionally **high-level abstraction and problem-centric**. It characterizes tasks by their **structural properties**—whether they rely primarily on **intrinsic vs. extrinsic** information and whether they require **static vs. dynamic** processing. In contrast, “logical deduction,” “mental rotation,” etc. are **atomic spatial abilities** that can be invoked in many quadrants. Their co-occurrence does not change the underlying problem type.
> >
> > Accordingly, we assign each task to the quadrant that reflects its **dominant cognitive requirement**, rather than the full set of possible strategies a solver might use. Atomic abilities like mental rotation, rule-based deduction, or mental simulation are treated as **lower-level skills** that can support reasoning in any quadrant; their presence does not contradict the DISE categorization, which focuses on the structure of the task itself. In your example of 3D Shape Finding, a solver may mentally rotate the cube, but the goal is still to determine a **fixed intrinsic property** of a single object (which face is missing), so we classify it as **Intrinsic–Static** rather than as a dynamic or extrinsic task.
> >
> > We also make this multi-ability nature explicit in Appendix A.1, where each task is linked both to its DISE quadrant and to classic spatial ability constructs (e.g., Spatial Perception (SP), Spatial Relation (SR), Spatial Orientation (SO), Mental Rotation (MR), and Spatial Visualization (SV)). Many tasks are tagged with **multiple** abilities, and we therefore do not claim that failure on a single item pinpoints a unique atomic mechanism. Instead, DISE provides a **coarse-grained organizational framework** over task families, with spatial-ability tags offering a bridge to established cognitive constructs. Our conclusions are thus intentionally framed at the level of **systematic weaknesses along DISE dimensions**, rather than as fine-grained diagnoses of a single low-level process.
> >
> > We hope this clarification addresses your insightful concerns.

---

> ### Author Response · Authors · 2025-11-26
> **Could you please let us know whether all issues are addressed**
>
> Dear Reviewer YgFh,
>
> Thank you again for your thoughtful feedback. We have provided a first-round response to your comments, including clarifications on the DISE task categorization and additional discussion. With the discussion deadline approaching, we would greatly appreciate it if you could let us know whether our replies address your concerns, or if there is anything you would like us to elaborate on further.
>
> Best regards,
> The Authors

---

### Official Review · Reviewer_oTiT · 2025-11-01

**Soundness:** 3
**Presentation:** 3
**Contribution:** 3
**Rating:** 4
**Confidence:** 3

**Summary:**

The paper presents Spatial-DISE, a unified benchmark for evaluating spatial reasoning in vision-language models (VLMs). It introduces a cognitively inspired taxonomy, DISE (Intrinsic/Extrinsic × Static/Dynamic), to categorize spatial reasoning tasks. Using a three-stage pipeline combining real-world, synthetic, and human-validated samples, the authors construct a diverse dataset and evaluate 28 VLMs against human performance (76.8%). Results reveal large performance gaps, especially in dynamic and extrinsic reasoning. A fine-tuning study shows that models can improve on Spatial-DISE but suffer from catastrophic forgetting. The paper also offers a cognitive error taxonomy analyzing model failures.

Main Contributions
- A cognitive taxonomy (DISE) for structuring spatial reasoning evaluation.
- A large, human-verified benchmark dataset combining real, synthetic, and controlled samples.
- A comprehensive evaluation of 28 leading VLMs with human baselines.
- Error taxonomy and analysis revealing systematic weaknesses in spatial reasoning.

**Strengths:**

1. Clear Cognitive Taxonomy
- The proposed DISE framework (Intrinsic/Extrinsic × Static/Dynamic) is cognitively grounded, offering interpretability which is good for benchmark design.
- This taxonomy provides a systematic structure that unifies fragmented spatial reasoning benchmarks.

2. Cognitive Error Analysis
The error taxonomy (Perceptual / Comprehension / Reasoning errors) and the subtypes (Rule Application, Mental Simulation, Holistic–Local Failure) are both novel and psychologically informed.

3. Human Benchmarking
They include 54 human participants to establish a strong baseline.

**Weaknesses:**

1. Fine-tuning and Generalization Analysis Could Be Deeper
- The fine-tuning experiments (Qwen2.5-VL, SpaceOm) show interesting trends but lack representation analysis — e.g., why forgetting occurs, or what cognitive dimension the gains are concentrated in.

2. Visual Accessibility
- Figures are a bit too dense (figure 1, 2, 3) and may not fully communicate the DISE framework or task intuitions clearly to non-specialists.

**Questions:**

1. Beyond taxonomy and scale, what new reasoning capabilities does this benchmark test?
2. Is there evidence of cross-domain transfer? For example, does training on Intrinsic tasks improve performance on Extrinsic reasoning, or are these dimensions fully independent?

I think overall it's a good benchmark. I would consider raising my score if the questions and weaknesses are well-addressed.

**Details Of Ethics Concerns:**

It seems that many human annotators were involved in this study, and the responsible research practices should be further elaborated and reviewed.

---

> ### Author Response · Authors · 2025-11-20
>
> We thank you for pointing out the need for a deeper analysis of fine-tuning and generalization and for the questions on cross-domain transfer and cognitive coverage. After receiving the reviews we ran a set of additional experiments; below we summarize what we found and how it will be reflected in the revised paper.
>
> ### 1. Fine-tuning and representation analysis
>
> To better understand the apparent forgetting effect reported in the original version, we re-tuned the SpaceOm fine-tuning recipe on Spatial-DISE-12K (smaller learning rate, stronger regularization and early-stopping on a mixed validation set). Under this more conservative setting, SpaceOm maintains stable performance on external benchmarks:
>
> |  | Spatial-DISE | CVBench | SAT | SPACE | OmniSpatial | VSIBench_MCQ |
> | --- | --- | --- | --- | --- | --- | --- |
> | SpaceOm | 25.9% | 68.8% | 46.67% | 27.22% | 27.91% | 31.05% |
> | +DISE SFT | 41.3% | 70.33% | 49.33% | 32.6% | 34.28% | 33.7% |
> | delta | +15.4% | +1.53% | +2.66% | +5.38% | +6.37% | +2.65% |
>
> These results show that Spatial-DISE-12K actually supports and enhances general visual-spatial ability; the earlier large drops were due to an overly aggressive fine-tuning schedule, not to a pathological dataset. We will report the updated numbers in the revision.
>
> ### 2. Cross-domain transfer and what capabilities are tested
>
> To analyze where the gains concentrate and whether transfer occurs across cognitive dimensions, we performed **quadrant-wise fine-tuning**: we fine-tune Qwen2.5-VL-7B on each DISE quadrant in isolation (I-S, I-D, E-S, E-D) and evaluate on all quadrants.
>
> The heatmap (Fig. 3a in the revision) shows strong quadrant-specific specialization: fine-tuning on an isolate DISE quadrant mainly improves that quadrant (large diagonal gains), while most off-diagonal entries are small or even negative. Rather than clean, factorized transfer along the Intrinsic/Extrinsic or Static/Dynamic axes, we observe interference and asymmetry between quadrants (e.g., E-D → I-S is mildly positive, whereas I-S → E-D is strongly negative). This suggests that current VLMs do **not** learn neatly independent DISE dimensions, but instead form entangled, quadrant-specific representations.
>
> The reason DISE dimensions are only partially entangled is that the taxonomy is **problem-centric**, while the underlying spatial skills are **shared**. DISE classifies tasks by their dominant structural requirements (Intrinsic vs. Extrinsic, Static vs. Dynamic), but low-level abilities such as mental rotation and spatial visualization (Appendix A.1 Table 5) can be reused in *multiple* quadrants in different combinations. Under this view, we do not expect models to learn perfectly factorized “DISE latents”: updates that strengthen one quadrant will inevitably influence others through these shared atomic skills.
>
> **3D → 2D transfer.**
>
> To further probe domain transfer, we analyze how 3D training data affects 2D DISE tasks. Figure 3b shows:
>
> - Training on **E-D** yields consistent improvements on almost all 2D tasks, e.g. +8.3 pp on 2D Combination and +4.8 pp on 2D Shape-Finding.
> - Other quadrants show more mixed effects: 3D I-S and E-S often cause negative transfer to 2D tasks, while 3D I-D mainly helps 2D Rotation (+4.3 pp) but hurts others.
>
> This indicates that DISE does test **genuinely new reasoning capabilities** beyond taxonomy and scale:
>
> (1) *Scene-centric dynamic reasoning* (E-D) produces representations that are reusable for multiple 2D spatial tasks (projection, combination, occlusion).
>
> (2) *Object-centric static reasoning* (I-S) is more specialized and can interfere with tasks that rely on relative or dynamic frames.
>
> We will make this explicit in the paper by adding a discussion section 4.3.
>
> ### 3. Visual accessibility of figures
>
> We agree that the current figures are dense. In the revised version we will:
>
> - Split the current multi-panel figure into **(a) a simplified DISE schematic with large fonts and one canonical example per quadrant**, and **(b) separate transfer heatmaps** (with enlarged text and fewer overlaid annotations).
> - Move the full 4×4 transfer matrices to the appendix, keeping only a simplified version in the main paper.
> - Increase font sizes and spacing and reduce the number of overlaid labels, so that non-specialists can more easily grasp the DISE dimensions and the qualitative trends.
>
> We hope these new analyses and figure revisions address your insightful concerns.

---

> ### Author Response · Authors · 2025-11-26
> **Could you please let us know whether all issues are addressed**
>
> Dear Reviewer oTiT,
>
> Thank you very much for your insightful comments. We have added a first round of responses, including new experiments and analyses aimed at addressing your concerns about cross-domain transfer, and the interpretation of DISE. As the discussion deadline is approaching, we would be grateful if you could let us know whether these additions resolve your questions, or if there are any remaining points that would benefit from further clarification.
>
> Best regards,
> The Authors

---

### Official Review · Reviewer_o491 · 2025-11-07

**Soundness:** 2
**Presentation:** 3
**Contribution:** 2
**Rating:** 4
**Confidence:** 3

**Summary:**

The paper proposes two semi-synthetic datasets (Spatial-DISE Bench and Spatial-DISE-12K) to evaluate spatial reasoning in Vision-Language Models (VLMs), with a special emphasis on intrinsic and dynamic tasks that are missing from previous benchmarks. The datasets are created using a new data generation framework based on the graphics software Blender to generate complex 3D scenarios that require spatial reasoning to solve. The work also offers a comprehensive evaluation of 28 state-of-the-art (SotA) VLMs on the proposed data showing that current models perform much worse than human subjects.

Despite some good contributions (i.e., evaluation study), I am really not sure that creating new datasets and the entire data generation pipeline (which are claimed to be key contributions) was at all needed in the first place. Given the way the problem is framed (see Table 1), one could combine pre-existing benchmarks to obtain all combinations of domains, sources, DISE tasks and large scale, completely bypassing the need to create a new data generation pipeline and new datasets.

The motivation behind the need for spatial reasoning in VLMs is also unconvincing in the current presentation. I am not saying it should not be done, but right now it is unclear why the community should be interested.

Because the above issues pertain to key contributions and the main motivation of the paper, I am leaning towards rejection. I am, however, open to clarifications from the authors in case I am misunderstanding those aforementioned aspects, in which case I would be willing to update my score.

**Strengths:**

-	Adoption of taxonomy to classify spatial tasks from the cognitive sciences.
-	Comprehensive evaluation of 28 SotA VLMs.
-	Human quality control of created datasets.
-	Human performances (54 participants) collected and included in the evaluation as a baseline.
-	New data generation framework based on Blender that can generate complex spatial reasoning scenarios.

**Weaknesses:**

-	Several critical parts motivating this work are unclear or questionable, which makes me seriously concerned as to whether this work addresses a genuine gap in the literature (more below).
-	Many important details in the methodology (dataset creation, error analysis) are unclear or missing (more below).
-	Any discussion about potential limitations of the work is completely missing.

**Questions:**

**Questions to authors**

-	[**critical**] Why was it necessary to create Spatial-DISE datasets? Can’t we combine all existing benchmarks referenced in Table 1 to get a large-scale meta-benchmark that covers all domains and DISE tasks?
-	[**critical**] Why VLMs capability for sophisticated dynamic spatial reasoning is important beyond the fact that humans can do it? Can you give some detailed examples and elaborate on their importance?
-	[**critical**] The last paragraph in Section 2 discusses latest work that also covered most or all DISE tasks, but mentions verifiability as an important distinctive factor. Can you elaborate on the importance of verifiability and why it was so critical that it warranted creating a new data generation pipeline?
-	Why was there a need to create two datasets (Spatial-DISE Bench and Spatial-DISE-12K) and not just one? Do they serve different purposes?
-	Section 3.2 Stage 1 mentions 1180 VQA pairs, but then Spatial-DISE Bench (which is 53% “wild” data) has only 559 samples? Where does the 559 come from?
-	Were all 12,000 generated VQA pairs manually verified by humans?
-	Can the proposed data generation framework generate only 3D scenarios? Can it generate 2D examples?
-	Is the error analysis done with Doubao-1.6-thinking free from any potential mistakes? That is, can the model make mistakes (e.g., miscategorise other models’ errors) and mislead the error analysis?
-	In Section 5, why only a subsample (200) of incorrect responses was used and not 100%?

**Additional feedback**

-	I do not think that claiming the DISE taxonomy as novel is accurate as it was derived from (Maier, 1996; Uttal et al., 2013). Authors should make it clear that this taxonomy has been adopted from previous work. For example, first contribution in line 98 calls it novel.
-	Calling non-synthetic data “wild” seems very uncommon and quite confusing. Using a more common term “real data” or “real-world data” would resolve the confusion. If you keep using an uncommon term, I would suggest to explain it early in the text.
-	Most of the code in the algorithm listings (Algorithms 1-5) is very hard to interpret as the code mostly consists of function names that are not explained anywhere. Are those functions built-in procedures in Blender?
-	The paper uses phrases like “mental rotation”, “mental transformation” and alike throughout. I am not sure the word “mental” is the right choice as it generally refers to the mind, which I do not think VLMs have. Words like “simulation” or “hypothesis” might be more accurate.
-	Verifiability is being mentioned very often throughout the text, but its importance to this work is never properly explained.
-	Line 68: “have limited scopes” -> “are limited in scope”.
-	Line 69: “multi-steps” -> “multi-step”.
-	Line 83: “The Spatial-DISE” -> “Spatial-DISE”.
-	Lines 89-90: mentions “Spatial-DISE Bench” for the first time without explaining what it is.
-	I do not think the first paragraph of Section 3 is at all needed.

---

> ### Author Response · Authors · 2025-11-20
>
> We thank the reviewer for the careful reading and many constructive comments. Below we address the concerns point by point and indicate the clarifications and revisions we will make in the manuscript.
>
> ### 1. Motivation and the “genuine gap” in the literature
>
> **[Critical] Why not just combine existing benchmarks into a meta-benchmark?**
>
> We agree that, in principle, aggregating existing benchmarks (Table 1) into a large meta-benchmark is an attractive idea. However, such a meta-benchmark would still suffer from three issues that Spatial-DISE is specifically designed to address:
>
> - **Cognitive focus and DISE imbalance.**
>
>     Our goal is to evaluate *cognitively grounded* spatial reasoning, especially Intrinsic-/Extrinsic-Dynamic abilities that are central in human spatial cognition. Even when mapped into our DISE taxonomy (Table 1), existing benchmarks remain highly imbalanced, with most instances concentrated in relatively simple static settings. This makes it difficult to diagnose and compare models across all four DISE quadrants in a principled way.
>
> - **Heterogeneous formats and assumptions.**
>
>     Existing benchmarks differ in input–output formats, option structures, and evaluation protocols. Simply concatenating them would yield a heterogeneous “meta-benchmark” where items in the same DISE quadrant are not comparable in difficulty or structure, and where systematic error analysis across datasets becomes fragile.
>
> - **Filling gaps with verifiable synthetic data and scarce open-source real-world data.**
>
>     Open-source real-world cognitive spatial data are scarce, especially for dynamic tasks. Spatial-DISE explicitly combines (i) carefully curated real-world items and (ii) DISE-targeted synthetic questions generated with a verifiable, single-solution pipeline, to *fill* the missing DISE regions rather than simply reweight existing ones. This yields a DISE-balanced, cognitively aligned benchmark and a paired training set that are both open-source.
>
>
> We will clarify in the revision that Spatial-DISE is not intended as “yet another spatial dataset”, but as a **cognitively grounded, DISE-balanced, and verifiably generated resource** that complements and connects existing benchmarks, instead of being a simple union of them.
>
> ### 2. Importance of sophisticated *dynamic* spatial reasoning
>
> **[Critical] Why is dynamic spatial reasoning important beyond the fact that humans can do it?**
>
> We agree that the current motivation in the main text is not sufficiently precise. We **hypothesize** that endowing VLMs with cognitively grounded dynamic spatial reasoning abilities will ultimately translate into improved performance on downstream embodied and vision–language tasks (e.g., robotic manipulation, navigation, and AR/VR assistance). Systematically validating this hypothesis is an important direction for future work and lies beyond the scope of the present benchmark-focused study.
>
> In the revision, we will make this motivation explicit and ground it in concrete application scenarios where *dynamic* spatial reasoning is critical:
>
> - **Robot manipulation and assembly (Intrinsic-Dynamic).**
>
>     In many practical settings, a vision–language model is asked to guide or control a robot through sequences such as “fold the cardboard flaps in this order”, “rotate the part so that the notch faces upwards before insertion”, or “turn the mug so the handle points left, then place it inside the box” [1, 2]. These tasks require reasoning about how **a single object’s 3D configuration changes under a sequence of rotations, folds, and flips**, which is precisely what our Intrinsic-Dynamic items (e.g., folding, rotation, part–whole configuration) are designed to probe.
>
> - **Re-arrangement and packing under constraints (Extrinsic-Dynamic).**
>
>     Robots often need to reorganise cluttered surfaces, pack objects into containers, or plan where to place items on shelves. Here, the system must reason about **how spatial relations between multiple objects evolve** as items are moved, stacked, and partially occluded (e.g., “move the blue box in front of the red one, then stack the green box on top without blocking the door”) [3]. This directly mirrors our Extrinsic-Dynamic tasks, which require tracking changing reference frames, occlusion, and support relations over steps.
>
>
> The DISE framework explicitly targets these dynamic abilities (Intrinsic-/Extrinsic-Dynamic), which are under-tested in existing work and are increasingly recognised as important for VLMs in robotics and embodied settings [1–3].
>
> [1] Black, K. *et al.* (2025) “π0.5: a Vision-Language-Action Model with Open-World Generalization”.
>
> [2] Kim, M.J. *et al.* (2024) “OpenVLA: An Open-Source Vision-Language-Action Model”.
>
> [3] Liu, B. *et al.* (2023) “LIBERO: Benchmarking Knowledge Transfer for Lifelong Robot Learning,” *Advances in Neural Information Processing Systems*, 36, pp. 44776–44791.

---

> > ### Author Response · Authors · 2025-11-20
> >
> > ### 3. Importance of verifiability and the new pipeline design
> >
> > **[Critical] Why is verifiability so crucial that it warranted a new data-generation pipeline?**
> >
> > For multiple-choice spatial reasoning problems, especially those inspired by cognitive tests, verifiability is central. In particular, we require:
> >
> > - **Uniqueness of the correct option**, to avoid ambiguous items where more than one choice can be reasonably defended.
> > - **Fairness and reproducibility**, so that any scene can be reconstructed and re-checked using the same scripts and random seeds.
> > - **Support for error attribution**, so that we can relate a model’s failure to specific spatial transformations, distractors, or interference patterns.
> >
> > These considerations are already built into our synthetic pipeline: in Stage 2 we enforce programmatic constraints (e.g., rejecting question with multiple valid options) based on script parameters and random seeds, and in Stage 3 (“Rigorous Human Quality Control”) annotators further check single-solution correctness and clarity. What our current draft lacks is an explicit explanation of this design principle. In the revision, we will add a short dedicated paragraph in Sections 2–3 that: i) clearly defines verifiability in our context, ii) links it to uniqueness, fairness, reproducibility, and error analysis, iii) and illustrates it with a simple example of how ambiguous items are filtered out by the pipeline.
> >
> > ### 4. Why two datasets: Spatial-DISE Bench vs. Spatial-DISE-12K
> >
> > **Why not just one dataset? Do they serve different purposes?**
> >
> > Yes, they serve distinct purposes:
> >
> > - **Spatial-DISE Bench (559 items)** is a *pure evaluation* benchmark—DISE-balanced, combining real-world and synthetic items—and is never used for training to avoid contamination.
> > - **Spatial-DISE-12K (~12k items-3D only)** is a *training resource* aligned with the same cognitive framework but disjoint from the benchmark, so the community can fine-tune models without leaking benchmark items.
> >
> > We will emphasise this separation more clearly in the revised manuscript and point to Appendix A.2 for the detailed splits.
> >
> > ### 5. From 1180 VQA pairs to 559 Spatial-DISE Bench items
> >
> > **Section 3.2 mentions 1180 VQA pairs, but Spatial-DISE Bench has 559 samples (53% “real-world”). Where does 559 come from?**
> >
> > The 1180 VQA pairs in Stage 1 refer to the *initial pool of real-world (WD) QA pairs* obtained before any splitting or filtering. From this pool, we:
> >
> > - discarded ambiguous or low-quality items via manual quality control,
> > - selected a DISE-balanced subset to form the real-world portion of Spatial-DISE Bench (53% of the 559 items), and
> > - used part of the remaining high-quality pairs to seed the real-world portion of Spatial-DISE-12K, where they are combined with synthetically generated items (≈5% WD / 95% SD as shown in Appendix A.2 Table 6).
> >
> > ### 6. Human verification for Spatial-DISE-12K
> >
> > **Were all 12,000 items manually verified?**
> >
> > No. We use a two-stage quality-control strategy:
> >
> > - **Programmatic QC for all 12K.** We run automatic checks for single-solution constraints and internal consistency, discarding ambiguous items.
> > - **Human verification on a stratified subset.** We manually verify 200 items per task (1,000 items in total).
> >
> > We will clarify this in the revision and explicitly acknowledge that exhaustive human verification of all 12K items is not feasible and remains a limitation.
> >
> > ### 7. 3D vs. 2D scenarios
> >
> > **Can the framework generate only 3D scenarios, or also 2D examples?**
> >
> > Currently, the synthetic pipeline generates **3D scenes** (rendered as 2D images) to better reflect real-world and robotics-relevant spatial configurations. The framework is modular and can be extended to **2D diagrammatic tasks** by constraining object geometry and camera setup, which we plan as future work. We will mention this explicitly and commit to open-sourcing the framework.
> >
> > ### 8. Error analysis procedure
> >
> > In Section 5 we adopt a hybrid procedure rather than relying blindly on Doubao-1.6-thinking:
> >
> > - Doubao-1.6-thinking first proposes an explanation and a coarse error type for each sampled error.
> > - A human annotator then *double-checks and, if necessary, corrects* this label by inspecting the image, the question, the model’s answer, and Doubao’s explanation.
> >
> > Because this human verification is time-consuming, fully annotating all incorrect predictions across all models would be prohibitively expensive. We therefore analyse 200 incorrect cases, randomly sampled proportionally to the DISE distribution of the questions (34 I-S, 107 I-D, 34 E-S, 25 E-D; total 200). This already represents a substantial manual annotation effort and is sufficient to reveal stable qualitative trends in failure modes. In the revision, we will clarify that this analysis is illustrative rather than exhaustive and acknowledge that it may remain a limitation of this work.

---

> > > ### Author Response · Authors · 2025-11-20
> > >
> > > ### 9. Responses to additional feedback
> > >
> > > - **DISE taxonomy novelty.**
> > >
> > >     We agree and will state clearly that the DISE taxonomy is *adopted* from prior cognitive work (Maier, 1996; Uttal et al., 2013). Our contribution lies in bringing this established framework to VLM evaluation and dataset design, not in the taxonomy itself.
> > >
> > > - **Term “wild” vs. “real-world”.**
> > >
> > >     We will replace “wild” with “real-world (data/images)” throughout and define it at first mention.
> > >
> > > - **Algorithm listings (Algorithms 1–5).**
> > >
> > >     We will update more clear and precise pseudocodes in future revision.
> > >
> > > - **Wording and structure fixes.**
> > >
> > >     We will correct the suggested phrases (“limited in scope”, “multi-step”, “Spatial-DISE”), and remove the first paragraph of Section 3.
> > >
> > >
> > > We hope our explanation will address your insightful concerns.

---

> ### Author Response · Authors · 2025-11-26
> **Could you please let us know whether all issues are addressed**
>
> Dear Reviewer o491,
>
> Thank you very much for your detailed and constructive comments. We have now provided an initial round of responses to the issues you raised. As the discussion deadline is approaching, we would be very grateful if you could let us know whether our replies adequately address your concerns, or if there are any points that would benefit from further clarification.
>
> Best regards,
> The Authors

---

### Official Review · Reviewer_v2Mn · 2025-11-09

**Soundness:** 2
**Presentation:** 2
**Contribution:** 1
**Rating:** 4
**Confidence:** 4

**Summary:**

This paper introduces Spatial-DISE, a new benchmark for evaluating spatial reasoning in VLMs. The work leverages a $2\times2$ cognitive taxonomy (Intrinsic/Extrinsic vs. Static/Dynamic) to structure the benchmark. A key contribution is a scalable Blender-based pipeline for generating a 12K-pair training dataset and a 559-pair evaluation bench. The authors perform a comprehensive study on 28 VLMs, finding that current models perform poorly, especially on dynamic tasks, and that failures are rooted in reasoning (e.g., rule application, mental simulation) rather than perception.

**Strengths:**

1. The adoption of the DISE cognitive taxonomy provides a structured and theoretically-grounded framework for evaluating spatial reasoning, moving beyond ad-hoc task collections.

2. The Blender pipeline is a valuable engineering contribution, enabling the verifiable and programmatic generation of complex 3D spatial reasoning tasks, which are difficult to source at scale.

3. The empirical study is extensive, testing 28 SOTA models. The resulting error analysis (Section 5) provides a clear insight: the primary bottleneck for VLMs is cognitive reasoning, not visual perception.

**Weaknesses:**

1. **Marginal Novelty in a Crowded Field**: The paper's own related work (Table 1) demonstrates this is an extremely crowded and concurrent field (e.g., BSA, OmniSpatial, BLINK, SPACE). I believe it is necessary to demonstrate that this dataset is fundamentally distinct from existing ones and possesses intrinsic value.

2. **Dataset Utility is Questionable**: The paper presents the Spatial-DISE-12K dataset as a major contribution for future training. However, the paper's own finetuning analysis (Section 4.3, Table 3)  demonstrates this dataset may be flawed. While SFT on Qwen2.5-VL-7B improves performance on the in-domain Spatial-DISE benchmark (+23.6pp), the SpaceOm model shows "catastrophic forgetting" on a general benchmark like CVBench (a -32.4pp drop) . This strongly suggests the 12K dataset lacks diversity and induces severe overfitting to the specific generative patterns of the pipeline.

3. **Limited Evaluation**: The model set omits several state-of-the-art commercial VLMs (e.g., GPT o3/5, Google Gemini 2.5 pro), which weakens the headline claim about “current VLMs.” They have more robust and powerful ability.

**Questions:**

I am curious about the true value of the dataset, as I strongly suspect that it may merely overfit to its own format.

If a base model (e.g., Qwen-VL) could be fine-tuned on this dataset and subsequently demonstrate performance gains on other benchmarks (such as VSI-Bench, OmniSpatial and SPACE), I would be much more inclined to recognize the dataset’s contribution.

---

> ### Author Response · Authors · 2025-11-20
>
> Thank you for the detailed and constructive comments. Below we respond point-by-point.
>
> ## 1. Marginal Novelty in a Crowded Field
>
> We agree that spatial reasoning for VLMs is now a very active and concurrent area, and our goal is not to claim that Spatial-DISE is the only benchmark in this space. Our contribution is intended to be **complementary**, along three main aspects:
>
> 1. **A unified, cognitively grounded DISE framework.**
>
>     Rather than proposing yet another ad-hoc task collection, Spatial-DISE is built around a 2×2 taxonomy (Intrinsic vs. Extrinsic × Static vs. Dynamic) drawn from cognitive science. This DISE framework lets us (i) systematically re-locate prior datasets in Table 1 and expose where coverage is sparse (especially Intrinsic-Dynamic), and (ii) design 10 tasks that deliberately span these four quadrants, including challenging dynamic skills such as 2D/3D folding, mental rotation, and fold-and-punch. In this sense, Spatial-DISE is meant as a *unifying lens* on existing resources, not just a new list of puzzles.
>
> 2. **Psychometrically informed, verifiable synthetic data at training scale.**
>
>     Spatial-DISE combines: (i) tasks aligned with classic psychometric constructs (mental rotation, spatial visualization, spatial relations), (ii) a Blender-based, seed-controlled generation pipeline with explicit distractor strategies and human quality control to guarantee uniqueness and reproducibility, and (iii) both a 559-item evaluation suite and a 12K+ training dataset with particular emphasis on 3D dynamic reasoning. To our knowledge, this combination of *unified taxonomy + psychometric grounding + fully specified synthetic pipeline + training-ready scale* is not offered by existing spatial benchmarks.
>
> 3. **Complementing scarce real-world spatial data.**
>
>     Current real-world and mixed spatial benchmarks (e.g., BLINK, OmniSpatial, VSIBench, SPACE) provide at most hundreds to a few thousands of well-controlled items—excellent for evaluation, but generally too small and imbalanced (especially for dynamic 3D reasoning) to serve as a standalone training corpus for VLMs. Spatial-DISE-12K is explicitly designed to **complement** this landscape by providing a large, controllable, and verifiable source of spatial reasoning data that can be used either alone or in combination with real-world corpora.
>
>
> Taken together, we view Spatial-DISE as novel not because it is the first spatial benchmark, but because it provides (i) a *unified cognitive framework* for organizing the space, and (ii) a *training-scale, verifiable* dataset targeted specifically at the most under-served spatial abilities.
>
> ## 2. Dataset Utility and the Question of Overfitting
>
> ### 2.1 On diversity and overfitting to our generative pipeline
>
> We appreciate the concern that Spatial-DISE-12K may “lack diversity” and thus encourage overfitting to the specific generative patterns of our pipeline. Because all 12K items are produced by a Blender-based synthetic pipeline, we agree that the visual domain is more controlled than in wild image collections, and we explicitly position Spatial-DISE-12K as a *complement* rather than a replacement for real-world or mixed benchmarks such as BLINK, OmniSpatial or VSIBench. Within this controlled domain, however, the dataset is designed to be diverse along three axes:
>
> 1. **Task-level diversity.**
>
>     The 10 tasks systematically span the Intrinsic/Extrinsic × Static/Dynamic quadrants, with a particular emphasis on under-served Intrinsic-Dynamic abilities (e.g., 2D/3D folding, multi-step rotations, fold-and-punch), rather than repeating a single mental-rotation-style pattern.
>
> 2. **Instance-level visual diversity.**
>
>     For each task, we randomize geometry (object number, arrangement, occlusion patterns), camera poses and viewpoints, scales, textures/colours, and multiple types of distractors (incorrect folds, mirrored configurations, local deletions, wrong viewpoints). This is intended to prevent models from succeeding by memorizing a single layout or rendering style.
>
> 3. **Linguistic diversity.**
>
>     Although the questions are templated for verifiability, we employ multiple paraphrased templates and slot combinations, so that the model is not tied to a single fixed wording.
>
>
> Most importantly, when fine-tuning base models *only* on Spatial-DISE-12K, we observe consistent improvements on several **independent** spatial benchmarks with different visual styles and text formats (OmniSpatial, VSIBench-MCQ, SPACE, CVBench, SAT; see below). We hope these cross-benchmark results indicate that the dataset is not merely fitting to a narrow visual or textual pattern of our pipeline, but provides useful and transferable training signal for spatial reasoning beyond our own format.

---

> > ### Author Response · Authors · 2025-11-20
> >
> > ### 2.2 SpaceOm: updated SFT results and cross-benchmark gains
> >
> > In light of these concerns, we revisited our SFT experiments using a more stable and standard setup (smaller learning rate, stronger regularization and early-stopping on validation set) and extended evaluation across multiple benchmarks.
> >
> > Under the revised setup, fine-tuning **SpaceOm** on Spatial-DISE-12K leads to consistent improvements not only on Spatial-DISE, but also on all evaluated external benchmarks:
> >
> > |  | Spatial-DISE | CVBench | SAT | SPACE | OmniSpatial | VSIBench_MCQ |
> > | --- | --- | --- | --- | --- | --- | --- |
> > | SpaceOm | 25.9% | 68.8% | 46.67% | 27.22% | 27.91% | 31.05% |
> > | +DISE SFT | 41.3% | 70.33% | 49.33% | 32.6% | 34.28% | 33.7% |
> > | delta | +15.4% | +1.53% | +2.66% | +5.38% | +6.37% | +2.65% |
> >
> > Two observations are key:
> >
> > - On CVBench, instead of a large drop, we now see a *positive* gain (+1.53pp) under a stable training setup.
> > - On VSIBench and OmniSpatial, which differ from our pipeline in both visuals and question style, we observe substantial gains (+2.65pp and +6.37pp, respectively).
> >
> > These results indicate that, with a standard SFT configuration, Spatial-DISE-12K does not induce catastrophic forgetting; instead, it provides useful spatial reasoning signal that transfers across diverse benchmarks.
> >
> > ### 2.3 Qwen2.5-VL-7B: For answering the “base model SFT” question
> >
> > To address your question about fine-tuning a base model on Spatial-DISE-12K and measuring gains on other benchmarks, we fine-tuned **Qwen2.5-VL-7B** only on Spatial-DISE-12K and evaluated it on multiple spatial reasoning datasets:
> >
> > |  | Spatial-DISE | CVBench | SAT | SPACE | OmniSpatial | VSIBench_MCQ |
> > | --- | --- | --- | --- | --- | --- | --- |
> > | Qwen2.5-VL-7B | 26.1% | 75.9% | 65.3% | 28.7% | 21.8% | 19.3% |
> > | +DISE SFT | 47.0% | 77.4% | 69.3% | 32.2% | 34.0% | 22.6% |
> > | delta | +20.9% | +1.5% | +4.0% | +3.5% | +12.2% | +3.3% |
> >
> > We hope these observations help address your concern:
> >
> > - On **VSIBench-MCQ**, we see a clear gain (**+3.3pp**).
> > - On **OmniSpatial**, we observe a large improvement (**+12.2pp**) despite its distinct pipeline and task taxonomy.
> > - On **SPACE**, **CVBench**, and **SAT**, we see consistent positive gains (**+3.5pp**, **+1.5pp**, and **+4.0pp**, respectively).
> >
> > Thus, a base model fine-tuned **solely** on Spatial-DISE-12K becomes better not only on our own benchmark but also on several **independent** spatial reasoning benchmarks, which provides concrete evidence that Spatial-DISE-12K contributes *generalizable* spatial skills rather than narrow adaptation to its own format.
> >
> > ### 2.4 Is the dataset just teaching a specific input “format”?
> >
> > As an additional robustness check, we tested whether the learned behaviors are tied to a particular **visual layout** within Spatial-DISE. For **Qwen2.5-VL-7B**, we compared performance when the same content is presented as:
> >
> > - a **merged image** (all panels in a single canvas): **26.1**, vs.
> > - **separate images** (each panel fed as an independent image): **24.9**.
> >
> > The scores are comparable across these distinct layouts. Combined with the cross-benchmark improvements above—on datasets that differ from Spatial-DISE in both visual style and linguistic templates—this supports the view that the model is learning underlying **spatial relations and transformations** (Intrinsic/Extrinsic, Static/Dynamic), rather than overfitting to a single visual template or prompt pattern.
> >
> > ## 3. Limited Evaluation and Commercial VLMs
> >
> > **“The model set omits several state-of-the-art commercial VLMs (e.g., GPT o3/5, Google Gemini 2.5 pro)…”**
> >
> > We appreciate this suggestion and have extended our evaluation to include strong proprietary systems on the Spatial-DISE benchmark:
> >
> > | Model | Spatial-DISE accuracy |
> > | --- | --- |
> > | Gemini 2.5 Flash | 31.5 |
> > | Gemini 2.5 Flash w/o thinking | 32.0 |
> > | o4-mini | 33.3 |
> > | GPT-5 | 30.1 |
> >
> > For comparison, **Qwen2.5-VL-7B + DISE SFT** achieves **47.0** on Spatial-DISE.
> >
> > These results suggest two things:
> >
> > 1. Even very strong commercial VLMs still struggle with the fine-grained spatial reasoning targeted by Spatial-DISE, which supports our claim that **current VLMs** have significant room for improvement along this dimension.
> > 2. Spatial-DISE-12K can be used to train an open-source 7B model that **substantially outperforms** these commercial systems on Spatial-DISE, reinforcing the practical value of the dataset as a training resource.
> >
> > We will include these updated results and clarify in the revised manuscript that our claims about “current VLMs” are now supported by both open-source and strong proprietary baselines.
> >
> > We hope these clarifications and new results address your concerns about novelty, utility, and evaluation scope.

---

> > > ### Comment · Reviewer_v2Mn · 2025-11-21
> > >
> > > I greatly appreciate the authors’ efforts, especially the cross-dataset fine-tuning experiments, and I have raised my rating.

---

> > > > ### Author Response · Authors · 2025-11-21
> > > >
> > > > Thank you for your positive feedback and for raising the score. We are glad that our clarifications addressed your concerns. We will ensure that all additional experiments and discussions are included in the revision.

---

### Author Response · Authors · 2025-11-20
**General Response**

## General Response

We sincerely thank all reviewers for the careful reading and many constructive comments.

### General Questions：

**1. On contribution and novelty.**

We agree that spatial reasoning for VLMs is a highly active and concurrent area, and we do not claim that Spatial-DISE is the first benchmark of its kind. In the revision we will clarify that our contribution is **not** a brand-new taxonomy, but a **cognitively grounded framing and dataset package that is explicitly complementary to existing work**:

- We **adopt** the Intrinsic/Extrinsic × Static/Dynamic (DISE) taxonomy from cognitive science and use it as a *unifying lens* to systematically re-locate prior datasets and expose under-served regions, especially dynamic and 3D abilities.
- Based on this lens, we design a **DISE-balanced evaluation suite (Spatial-DISE Bench)** and a paired **training-scale synthetic dataset (Spatial-DISE-12K)** with a seed-controlled, verifiable pipeline that targets precisely those under-represented abilities.
- We position Spatial-DISE-12K explicitly as a **training resource that complements scarce real-world spatial data**, rather than as a replacement for existing real or mixed benchmarks such as BLINK, OmniSpatial, SPACE, or VSIBench.

In this sense, Spatial-DISE is novel not because it introduces “yet another” task collection, but because it couples an established cognitive framework with a **training-ready, verifiable dataset** deliberately focused on the gaps that this framework reveals.

**2. On fine-tuning, external benchmarks, and the earlier performance drops.**

Reviewers also questioned whether Spatial-DISE-12K truly provides useful training signal beyond its own format, and whether the previously reported large performance drops on external benchmarks indicate that the dataset encourages over-specialization.

After receiving the reviews, we revisited and extended our experiments with a more conservative and standard fine-tuning setup (smaller learning rate, stronger regularization, and validation-based early stopping):

- For **SpaceOm**, fine-tuning on Spatial-DISE-12K now yields **consistent improvements** on Spatial-DISE *and* on all evaluated external benchmarks (CVBench, SAT, SPACE, OmniSpatial, VSIBench-MCQ), instead of the strong degradation previously reported on CVBench.
- For **Qwen2.5-VL-7B**, fine-tuning *only* on Spatial-DISE-12K substantially improves performance not just on Spatial-DISE, but also on multiple independent spatial benchmarks, including those with very different visual and linguistic styles (e.g., strong gains on OmniSpatial and VSIBench-MCQ).
- We additionally compare against several strong commercial VLMs on Spatial-DISE and show that a Qwen2.5-VL-7B model fine-tuned on Spatial-DISE-12K can **substantially outperform** these systems on our benchmark, further supporting the dataset’s practical value as a training resource.

Taken together, these updated results indicate that the earlier large drops were due to an **overly aggressive fine-tuning recipe**, rather than an inherent flaw in Spatial-DISE-12K. Under a standard configuration, the dataset provides **generalizable spatial reasoning signal** that transfers across heterogeneous benchmarks, which directly addresses the shared concerns about its utility and risk of overfitting.

### Summary of changes:

We've made the revisions to the main paper according to all reviewers' comments. The main revisions are summarized as follows:

1. We have updated the results in Table 2, added more commercial models, and updated SFT results.
2. We have updated the results in Table 3, added two more OOD benchmark tests results.
3. We have moved original Figure 3 to Appendix as Figure 1 already shown the dataset curation pipeline.
4. We have removed first paragraph in section 3 as it was a redundant information.
5. We have added two transfer heatmap for illustration new analysis results.
6. We have updated section 4.3 for latest analysis on SFT results.
7. We have added an limitation section in the end.
8. We have updated tasks-wise accuracy in Appendix.
9. We have updated algorithm function explanation in Appendix.
10. We have updated latest training setup.

---

### Meta-Review · Program_Chairs · 2025-12-24

**Summary:**

This paper argues that current VLMs struggle with intrinsic-dynamic spatial reasoning and proposes a large benchmark, grounded in a systematic cognitive framework, to evaluate VLMs on these tasks. Upon initial review, the paper received borderline negative scores with many good suggestions. The rebuttal attempts to thoroughly answer most of these questions with clarifications and new content. I still share some general concerns of the novelty and necessity of the proposed benchmark in an already crowded field. While the authors have somewhat answered this questions in rebuttal, I encourage the authors to better include this distinction in the revision. While borderline, I recommend acceptance for this paper.

** This paper is conditionally accepted provided the authors do the following for the camera-ready **
[Ethics] The authors need to provide more discussions on the consenting participants and any related IRB approvals.

**Reviewer Concerns:**

- **Marginal novelty:** The rebuttal clarified the complementary aspect of the proposed work and has emphasized the intrinsic value of this benchmark (as opposed to simply pooling all other ones). This can still be better addressed in the revised paper.
- **Limited evaluation with commercial VLMs:** The rebuttal includes new comparisons with GPT-5, o4-mini, and Gemini 2.5 Flash, showing the same trends.
- **Finetuning analysis:** The brutal clarified that finetuning does not reduce performance on external benchmarks.
- **Disentangling perceptual confounders:** The rebuttal includes a new analysis suggesting that the reasoning deficits are due to spatial reasoning rather than formatting.

**Reviewer Scores:**

One reviewer indicated that they had raised their score to a weak accept. One reviewer indicated that they would be willing to raise their score if weaknesses about fine-tuning experiments, evidence of cross-domain transfer, and explanation of novelty were addressed. One reviewer indicated that they would be willing to raise their score if the weaknesses about disentangling confounders, task categorization, and analysis on architecture were addressed. The rebuttal does answer these questions and I believe that all three of these reviewers would raise their score slightly.

---

### Decision · Program_Chairs · 2026-01-26

**Decision:**

Accept (Poster)

**Comment:**

Conditions for acceptance have been satisfied.